# 🌀 ReVSI: Rebuilding Visual Spatial Intelligence Evaluation for Accurate Assessment of VLM 3D Reasoning

**Yiming Zhang** [*][1]  **Jiacheng Chen** [*][1]  **Jiaqi Tan** [1]  **Yongsen Mao** [2]  **Wenhu Chen** [3]  **Angel X. Chang** [1][4]

🌐 Project Page    GitHub    🤗 Hugging Face

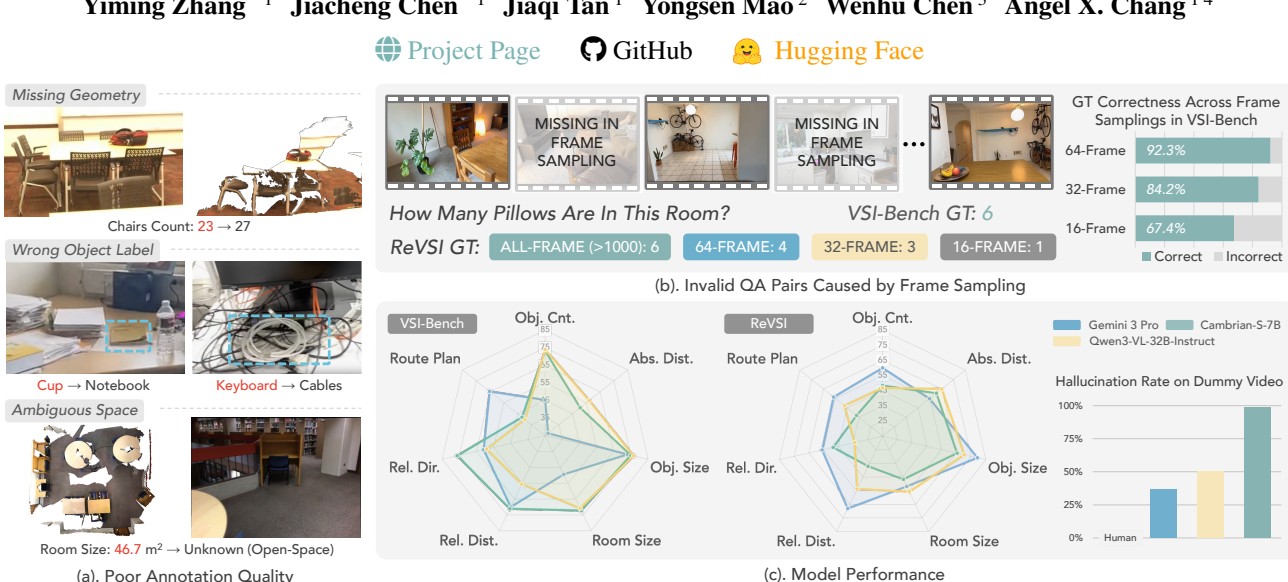

**(b). Invalid QA Pairs Caused by Frame Sampling**

**(a). Poor Annotation Quality**

**(c). Model Performance**

*Figure 1.* We revisit VSI-Bench (Yang et al., 2025a), a widely used benchmark for spatial reasoning, by systematically revealing issues in annotation correctness and bias (a) and pointing out that answers to questions should be sensitive to the input frames provided to the model (b). We develop a more accurate benchmark for assessing visual intelligence by thoroughly re-annotating, debiasing, and establishing new frame-aware protocols. We additionally construct *dummy-videos* by removing frames containing the queried objects, enabling controlled analysis of how models rely on visual evidence. Surprisingly, we find that proprietary models are under-assessed by VSI-Bench (*e.g.*, on object counting), while fine-tuned models show high hallucination rate under the dummy-videos setting (c).

## Abstract

Current evaluations of spatial intelligence can be systematically invalid under modern vision-language model (VLM) settings. First, many benchmarks derive question-answer (QA) pairs from point-cloud-based 3D annotations originally curated for traditional 3D perception. When such annotations are treated as ground truth for video-based evaluation, reconstruction and annotation artifacts can miss objects that are clearly visible in the video, mislabel object identities, or corrupt geometry-dependent answers (*e.g.*, size), yielding incorrect or am-
biguous QA pairs. Second, evaluations often assume full-scene access, while many VLMs operate on sparsely sampled frames (*e.g.*, 16–64), making many questions effectively unanswerable under the actual model inputs. We improve evaluation validity by introducing **ReVSI**, a benchmark and protocol that ensures each QA pair is answerable and correct under the model's actual inputs. To this end, we re-annotate objects and geometry across 381 scenes from 5 datasets to improve data quality, and regenerate all QA pairs with rigorous bias mitigation and human verification using professional 3D annotation tools. We further enhance evaluation controllability by providing variants across multiple frame budgets (16/32/64/all) and fine-grained object visibility metadata, enabling controlled diagnostic analyses. Evaluations of general and domain-specific VLMs on ReVSI reveal systematic failure modes that are obscured by prior benchmarks, yielding a more reliable and diagnostic assessment of spatial intelligence.

[*]Equal contribution  [1]Simon Fraser University  [2]Hong Kong University of Science and Technology  [3]University of Waterloo  [4]Alberta Machine Intelligence Institute (Amii).  Correspondence to: Yiming Zhang <yza440@sfu.ca>, Angel X. Chang <angelx@sfu.ca>.

*Proceedings of the 43rd International Conference on Machine Learning*, Seoul, South Korea. PMLR 306, 2026. Copyright 2026 by the author(s).

# 1. Introduction

As spatial reasoning becomes a key capability for vision-language models (VLMs), a growing body of benchmarks has emerged to evaluate visual spatial intelligence (VSI) in realistic 3D environments. However, we show that widely used benchmark designs and evaluation pipelines can be systematically invalid under modern VLM's video input settings, leading to unreliable conclusions.

In this paper, we identify two critical evaluation pitfalls behind this validity gap:

1) **Annotation-to-video ground-truth drift**. Repurposing point-cloud-based 3D annotations for video evaluation creates a systematic mismatch (see Figure 1(a) and Figure 8). Because these annotations are produced on imperfect reconstructions for traditional 3D perception, they can omit objects that are clearly visible in the raw video, assign incorrect object identities due to poor mesh quality, and corrupt geometry-dependent ground truth such as object and room size. As a result, a non-trivial fraction of derived QA pairs become incorrect or ambiguous under the video evidence, distorting the evaluation signal.

2) **Scene-observability mismatch.** Existing evaluations often implicitly assume full-scene observability, whereas practical VLM evaluation operates under strict input budgets and relies on sparsely sampled video frames. This mismatch makes key objects unobservable and can invalidate questions or their ground-truth answers under the actual model inputs, as shown by Figure 1(b).

To address these issues, we revisit and rebuild the widely used spatial intelligence benchmark, VSI-Bench (Yang et al., 2025a), with one guiding principle: making *what the model sees* strictly consistent with *what the benchmark asks*. Concretely, the proposed **ReVSI** benchmark:

- **Re-annotate** object labels and scene geometry using professional 3D annotation tools, improving annotation accuracy and diversity while making them fully consistent with the raw input videos.
- **Re-generate** QA pairs with more diverse templates and more rigorous answer-distribution control than prior work for bias mitigation, and applies careful human verification to ensure that each question is well-defined and supported by the input videos.
- **Re-define** frame-budgeted evaluation by providing variants at multiple frame budgets, enabling fair evaluation of models with different input budgets while ensuring that each setting yields valid and answerable QA pairs under the model's actual inputs.

Built on ReVSI's high-quality data and input-consistent evaluation pipeline, we obtain a more reliable picture of spatial understanding in modern VLMs. Crucially, by enforcing frame-level consistency between model inputs and benchmark QA, ReVSI enables diagnostic analyses that are not possible with prior benchmarks, where input QA mismatches confound whether failures arise from missing visual evidence or from deficient spatial reasoning. Moreover, the availability of fine-grained object visibility information supports a range of controlled experiments and diagnoses. We summarize several key findings below:

- **Base models (zero-shot).** ReVSI is more challenging due to its higher-fidelity and more diverse annotations. All evaluated proprietary models obtain stable or even higher performance on our improved benchmark, while open-source models exhibit substantial accuracy drops (up to 40%), especially on object counting, relative distance, and relative direction tasks (Table 3).
- **Finetuned specialized models.** Models finetuned on 3D spatial reasoning data show significantly smaller gains on ReVSI than on VSI-Bench. Moreover, scaling post-training data does not consistently improve performance, with some fine-tuned models performing worse on specific tasks than their base model (Table 4).
- **Visibility-based stress tests.** Leveraging frame-level object visibility, we construct evidence-absent controls, e.g., removing all frames containing the queried object for object counting (yielding a ground truth of zero), or replacing all frames with black images for object size estimation. While a grounded model should fail in these settings, several models (e.g., InternVL3.5 (Wang et al., 2025)) still achieve surprisingly high scores, revealing predictions driven by non-visual priors rather than visual evidence (Tables 5 and 6).

# 2. Related work

**Benchmarks for visual-spatial intelligence.** Early evaluations of spatial reasoning focus on models operating directly on 3D geometric data. Pioneering benchmarks (Azuma et al., 2021; Ma et al., 2022; Chen et al., 2020; Zhang et al., 2023) established the paradigm by grounding questions in 3D meshes from datasets such as ScanNet (Dai et al., 2017). As VLMs become increasingly capable, recent works have adapted this protocol to video-based and embodied settings. Benchmarks such as VSI-Bench (Yang et al., 2025a) and SPAR-Bench (Zhang et al., 2025a) assess spatial memory from video streams, while VSI-SUPER (Yang et al., 2025c) extends this to long-horizon 'supersensing' tasks like continual counting and tracking. Similarly, agent-centric suites like EmbSpatial-Bench (Du et al., 2024) and RefSpatial-Bench (Zhou et al., 2025) project annotations onto robot views. However, relying on existing 3D annotations for ground truth creates a validity gap, as projected annotations often misalign with the actual visual evidence in video frames, fundamentally undermining the reliability of the evaluation.

**Vision-language models for spatial reasoning.** State-of-the-art spatial reasoning is driven by generalist foundations and specialized post-training strategies. General-purpose models, including the GPT (OpenAI, 2025), Gemini (Gemini Team, 2023), and Qwen-VL / InternVL families (Qwen Team, 2025; Wang et al., 2025), demonstrate impressive zero-shot performance on standard benchmarks. To further specialize these capabilities, recent work employs targeted recipes: instruction-tuning approaches, such as SpatialVLM (Chen et al., 2024) and Cambrian-S (Yang et al., 2025c), align 2D perception with 3D concepts using massive synthetic data or curated spatial mixtures. Architecturally, models like VLM-3R (Fan et al., 2025) and Spatial-MLLM (Wu et al., 2025a) incorporate explicit geometry encoders or dual-stream modules to inject 3D cues into the visual representation. Reinforcement Learning methods like SpaceR (Ouyang et al., 2025) and 3D-R1 (Huang et al., 2025) optimize reasoning policies using geometry-consistent rewards. However, the validity of existing evaluations is often compromised by the systemic benchmark pitfalls identified in this work. When evaluated on ReVSI, we observe divergent conclusions and uncover new insights into the true spatial capabilities of these models.

## 3. Validity pitfalls in VSI evaluation

Visual-spatial intelligence benchmarks aim to measure spatial understanding of VLMs in realistic 3D environments. In this paper, we consider *evaluation validity* as the requirement that each QA pair is answerable and correct under the model's actual visual inputs. We show in Figure 1 that widely used evaluations can systematically violate this requirement, largely due to two root causes: insufficient *annotation accuracy* (ground truth and derived QA pairs drift from what is supported by the raw videos) and insufficient *controllability* where evaluations neither account for nor control the visual evidence available under realistic frame budgets. Below, we detail these two failure modes and quantify their impact.

**Annotation-to-video ground-truth drift.** Existing spatial intelligence benchmarks (Yang et al., 2025a; Zhang et al., 2025a) rely heavily on annotations derived from real-world scanned 3D indoor scene datasets (Dai et al., 2017; Baruch et al., 2021; Yeshwanth et al., 2023). However, such annotations often suffer from quality issues due to incomplete geometric reconstruction and the annotation workflows that operate on reconstructed meshes rather than raw videos. As a result, errors arise in object labels, sizes, positions, and even object existence (*i.e.*, missing or spurious objects). To assess the practical impact of these issues, we examine three representative subtasks from VSI-Bench: *object counting*, *room size and object size estimation*, and manually inspect all associated QA pairs (Figure 2). While we only focus on these three tasks due to the high cost

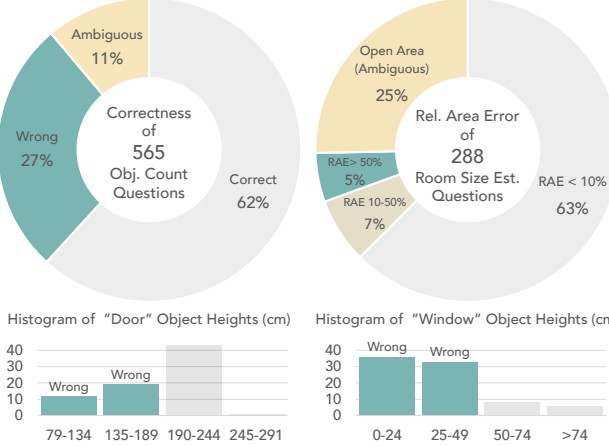

*Figure 2.* Error analysis on three representative VSI-Bench tasks. **(Top-Left)** Object counting shows notable incorrectness and ambiguity (*e.g.*, ill-defined object criteria like shoes). **(Top-Right)** Relative Area Error (RAE) for room size estimation, measured against human annotations, reveals frequent errors caused by ambiguous open-area scenes (*e.g.*, libraries) and noisy 3D reconstructions. **(Bottom)** Distributions of "door" and "window" heights as proxies highlight systematic object size errors, where bars labeled "Wrong" indicate physically implausible sizes.

of manual verification, the systemic absence of verification in prior benchmarks suggests that similar errors are pervasive across the entire dataset and other task types (Figure 8). These findings highlight a broader concern: when real-scanned indoor scene datasets are used for video-based evaluation, explicit verification and correction of annotations are essential to ensure ground-truth correctness and reliable evaluation.

**Scene observability under frame budgets.** Another often overlooked evaluation pitfall is that the visibility of queried objects in benchmark questions often varies with the video frame sampling rate. Prior evaluations often rely on intuition to select video frames, lacking a systematic study of object visibility. To quantify this effect, we analyze how the answerability and correctness of VSI-Bench questions change under different frame budgets. To isolate visibility effects from annotation noise, we assume all questions and ground-truth answers provided by VSI-Bench are valid under the all-frame setting, where the complete scene is visible. Frames are sampled uniformly across each video, following common practice in modern VLMs. As detailed in Figure 3 and Table 7, reducing the sampling rate substantially degrades evaluation validity, particularly when fewer than 32 frames are used, a setting commonly adopted in VLM fine-tuning and benchmarking (Ouyang et al., 2025; Wu et al., 2025b; Guo et al., 2025; Tang et al., 2025). Notably, Appearance Order questions remain severely impacted by object absence, resulting in persistent inaccuracies even with 64 frames. The above findings highlight frame sampling density as a pivotal factor. For real-scanned indoor scene datasets (Dai et al., 2017; Yeshwanth et al.,

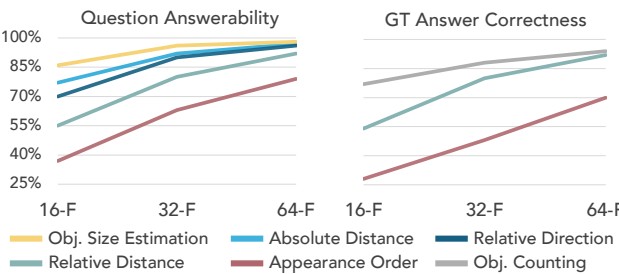

*Figure 3.* Answerability and correctness of VSI-Bench questions under different video sampling (16/32/64) settings. Questions become unanswerable when queried objects are absent, and incorrect when frame-sampled answers deviate from all-frame GT.

2023; Baruch et al., 2021) which primarily feature single-room environments, we recommend using a minimum of 64 uniformly sampled frames to ensure adequate visibility of scene elements.

## 4. Video-aligned annotation and QA curation

High-quality data is the cornerstone of reliable benchmarking. In this section, we detail our protocols for ensuring precise scene and object annotations, as well as our methodology for generating robust benchmark questions and answers under different video frame samples.

### 4.1. Video-aligned object and geometry reannotation

In contrast to prior benchmarks such as VSI-Bench (Yang et al., 2025a), which rely heavily on the low-quality original annotations of 3D scene datasets, we construct a comprehensive set of high-quality manual annotations. We developed a 3D web interface (Appendix B.1) and re-annotated object labels and 3D bounding boxes across ScanNetv2 (Dai et al., 2017), ScanNet++ (Yeshwanth et al., 2023), ARKitScenes (Baruch et al., 2021), 3RScan (Wald et al., 2019), and MultiScan (Mao et al., 2022). Using the original annotations as a starting point, we filtered out incorrect labels, refined inaccurate bounding boxes, and added new object instances.

We adopt an open-vocabulary setting for object labeling, with all labels manually annotated by humans and GPT-5.2 (OpenAI, 2025) used only for verification, enabling fine-grained and precise descriptions (*e.g.*, Sony PlayStation, Coca-Cola box). We ensure tight and correct 3D bounding boxes for all objects. In cases where object geometry is missing or fragmented, we extrapolate boxes to their physical dimensions by cross-referencing raw video frames and spatial context. To guarantee data quality, all annotation tasks were performed directly by authors with expertise in 3D datasets. Consequently, we notably expand the scale and diversity of object annotations compared to VSI-Bench, offering more scenes, more objects, and a richer open-vocabulary label set (Table 1 and Appendix C).

*Table 1.* Comparison of dataset statistics, highlighting larger scale and open-vocabulary support in ReVSI.

|  | Scenes | Objects | Obj. Labels | Open-Vocab |
|---|---|---|---|---|
| VSI-Bench | 288 | 3185 | 65 | |
| **ReVSI** | **381** | **5365** | **504** | ✓ |

### 4.2. QA re-generation with verification and bias control

To address the error rates observed in VSI-Bench (Yang et al., 2025a), we completely rebuild its question-answer pairs. While we adhere to the original task definitions and question formats, we refine the template-based generation strategy with stricter rules and enforced comprehensive human verification for every sample.

Figure 4 summarizes our newly constructed QA data. We exclude the *Object Appearance Order* task from VSI-Bench, as it primarily assesses temporal reasoning rather than 3D spatial intelligence and often involves partially visible or boundary objects, leading to ambiguous definitions of "appearance." For all remaining tasks, we refine and extend VSI-Bench generation heuristics as described below.

**Object counting.** The task counts instances of specified object categories within a scene. To avoid trivial cases, VSI-Bench excludes single-instance queries. However, predicting "2" alone achieves 62% accuracy, revealing strong dataset and environment bias (Figure 5). To prevent models from exploiting the bias and bypassing visual reasoning, we introduce two changes: (1) reintroducing single-count queries for common, frequently co-occurring categories (e.g., nightstands, pillows), which do not diminish the task difficulty since distinguishing repeated observations of the same object in long videos still requires 3D spatial reasoning; and (2) adding cumulative counting questions that aggregate instances across two object categories (Figure 4). Together, these changes significantly reduce the statistical bias observed in VSI-Bench.

We further revise the query template from *"How many objects are in this room?"* to *"How many objects are in the scene?"*; since many videos span multiple connected rooms or open-plan areas, VSI-Bench's original phrasing is misaligned with its ground truth, which often counts objects globally. Finally, we excluded object categories prone to counting ambiguity, such as shoes (Appendix B.7).

**Object size estimation.** This task estimates the longest dimension of an object in meters. In VSI-Bench, models can often rely on category-level priors rather than visual evidence. To reduce the bias, we first exclude object categories with near-fixed dimensions (*e.g.*, toilet and bed, which typically measure ∼2m). For other categories whose dimensions cluster around common ranges (*e.g.*, refrigerators), we selectively sample instances to ensure sufficient out-of-distribution examples (Appendix B.7). We

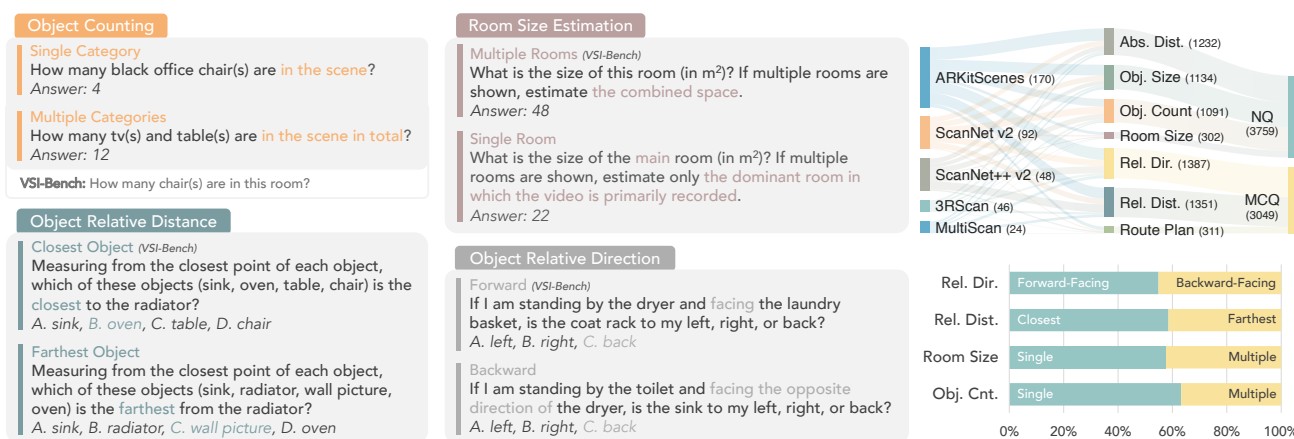

*Figure 4.* Compared to VSI-Bench, we introduce a richer and more balanced set of question templates, including cross-category object counting, relative distance for the farthest objects, avoiding ambiguous room estimation, and specifying different orientations with respect to an object (see Table 9 for details). On the right, we show the different question types, including the detailed ratios of variants, covering both numerical (NQ) and multiple-choice questions (MCQ), and sourced from 5 scene datasets (2 more than VSI-Bench).

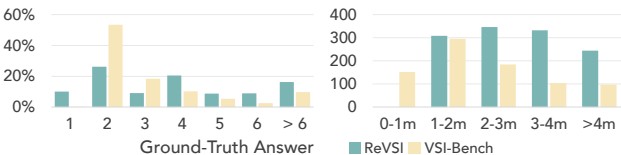

*Figure 5.* Answer distributions for object counting (left) and absolute distance (right). VSI-Bench is dominated by the answer "2" (53%) in object counting and by short ranges (0-2m) in absolute distance, where co-visible objects in a single frame favor 2D cues. In contrast, ReVSI shows more balanced counts and reduces short-range cases to emphasize long-range spatial reasoning.

further manually annotate 3D bounding boxes to obtain accurate object dimensions, substantially improving ground-truth quality over VSI-Bench, which derives approximate sizes from noisy 3D segmentation masks (Appendix B.3).

**Object absolute distance.** VSI-Bench includes many questions with ground-truth distances below 1m, which we argue are weak indicators of 3D spatial perception, as the queried objects are often co-visible in a single frame and solvable using 2D cues. We therefore remove these cases and add more long-range object pairs with diverse separations. As shown in Figure 5, ReVSI yields broader distance distributions, emphasizing long-range spatial reasoning.

**Object relative direction.** The questions evaluate egocentric spatial reasoning by instructing the agent to stand at a positioning object, face an orienting object, and infer the relative direction of a querying object. In VSI-Bench, ambiguity arises when large positioning objects (*e.g.*, beds) fail to define a precise starting point. We exclude positioning objects with footprints exceeding 1 m$^2$ and require a minimum separation of 1 m between referenced objects. We further add templates where the agent faces away from the orienting object to increase diversity (Figure 4).

**Object relative distance.** VSI-Bench exclusively asks for

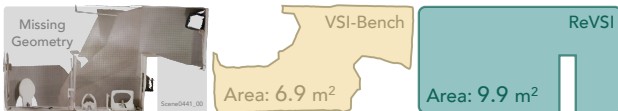

*Figure 6.* Comparison of room polygons of a bathroom between VSI-Bench and ReVSI. VSI-Bench computes the room area from the Alpha Shape (Edelsbrunner et al., 1983) algorithm on noisy reconstructed 3D geometry, while our data are manually re-annotated by referring to both 3D and the raw video.

the nearest object in relative distance questions. Following the original format, we introduce templates that query the farthest object to increase question diversity (Figure 4).

**Room size estimation.** This task estimates the total area of visible rooms in a video. To overcome limitations of VSI-Bench, which computes room areas using the Alpha Shape (Edelsbrunner et al., 1983) algorithm on noisy 3D reconstructions (Figure 6), we develop an annotation interface (Figure 10) and manually annotate room boundary polygons from orthogonal top-down views. Scenes with ill-defined boundaries are excluded. Moreover, VSI-Bench queries only the combined area of all rooms, which is ambiguous for partially captured open spaces. We therefore add an additional template asking for the area of the main single room, with corresponding ground-truth annotations.

## 5. Frame-aware evaluation

### 5.1. Frame-budgeted evaluation protocols

To enable evaluations under different video sampling budgets, we construct QA pairs for 16/32/64/all-frame sampling settings. For each sampling, we first rasterize the sampled frames using the ground-truth camera poses from scene datasets and compute 2D projections of visible objects. An object is considered visible if its pixel coverage

*Table 2.* Frame sampling consistency. ReVSI aligns annotation and inference frames and supports frame-adaptive question-answer pairs across sampling rates.

| | Annotation Frames | Inference Frames | Frame-Adaptive QAs |
|---|---|---|---|
| VSI-Bench | ALL | 8/16/32/64/128/ALL | |
| ReVSI | 16/32/64/ALL | 16/32/64/ALL | ✓ |

in the frame where it is most prominent exceeds 5% of the total frame area; otherwise, its visibility is manually annotated using a web interface (Figure 11). Moreover, we exclude the *Room Size Estimation* and *Route Planning* tasks from the 16-frame setting, as these questions are frequently unanswerable due to insufficient global scene context.

In contrast to VSI-Bench, which constructs data under the full-frame setting (often comprising thousands of frames) but evaluates models on subsampled inputs, we enforce both question answerability and ground-truth correctness under each sampling configuration. This substantially improves the benchmark quality and enables more reliable evaluation across different input budgets (Table 2).

### 5.2. Visibility-guided controlled diagnostics

To enable fine-grained diagnostics under controlled visual evidence, we construct a set of *dummy videos* under the 16-frame setting to probe models' sensitivity to visual input and their reliance on priors or hallucination.

For each scene, we start from the original video and remove all frames containing any object referenced by the associated question. The resulting video preserves scene context and all non-queried objects, while entirely excluding visual evidence of target objects (see Appendix B.5). These dummy videos are therefore unanswerable for humans: the queried objects never appear, and the ground-truth answer is deterministically defined (*e.g.*, zero for object counting).

This construction explicitly decouples scene context from task-relevant evidence. We use these dummy videos as controlled stress tests to assess whether model predictions are grounded in visual input or driven by memorized indoor-scene priors. As shown in Section 6, this diagnostic exposes systematic behavioral differences that remain hidden under standard evaluations, even when models achieve similar accuracy on real videos.

## 6. Experiments

With ReVSI, we enable a more reliable evaluation of models' spatial reasoning capabilities. We evaluate Qwen3-VL-Instruct (Qwen Team, 2025), InternVL-3.5 (Wang et al., 2025), and LLaVA-Video-Qwen2 (Zhang et al., 2025b) as representative open-source models, GPT-5.2 (OpenAI, 2025) and Gemini 3 (Google DeepMind, 2025) as represen-

tative proprietary models, together with specialized models (Yang et al., 2025c; Wu et al., 2025a; Fan et al., 2025; Ouyang et al., 2025; Yang et al., 2025b) fine-tuned for 3D spatial intelligence. Details are provided in Appendix D.

### 6.1. Evaluation setup

**Evaluation metrics.** We follow the metric design of VSI-Bench (Yang et al., 2025a). For Multiple-Choice Question (MCQ) tasks, we report exact-match *Accuracy* (*Acc*). For Numerical Question (NQ) tasks, we use *Mean Relative Accuracy* (*MRA*), which defines a prediction $\hat{y}$ as correct if the relative error falls within a dynamic threshold $\theta$. $MRA$ averages this performance across a set of confidence thresholds $\mathcal{C} = \{0.5, 0.55, \ldots, 0.95\}$:

$$MRA = \frac{1}{|\mathcal{C}|} \sum_{\theta \in \mathcal{C}} \mathbb{1}\left(\frac{|\hat{y} - y|}{y} < 1 - \theta\right), \qquad (1)$$

This metric penalizes large deviations while rewarding predictions that are relatively close to the ground-truth.

### 6.2. Results

**What flaws in VSI-Bench does ReVSI expose?** Table 3 reveals discrepancies between ReVSI and VSI-Bench in evaluating numerical reasoning. Across all numerical tasks, and particularly for object counting, VSI-Bench systematically underestimates the performance of proprietary models, making open-source models appear substantially stronger. This trend is reversed on ReVSI, where proprietary models consistently outperform open-source counterparts. Later, we show that this apparent advantage on VSI-Bench arises from severe hallucination behaviors commonly exhibited by open-source models, which are not adequately penalized under its biased evaluation data.

Notably, most models achieve higher scores on ReVSI for absolute distance estimation, despite ReVSI being more challenging due to the removal of short-range (<1m) questions and the inclusion of more long-range queries (Figure 5). We analyze this phenomenon in Figure 7 and attribute it to the interaction between metric design and model capability. Specifically, modern models such as Qwen3-VL (Qwen Team, 2025) demonstrate strong long-range distance estimation, with absolute error increasing noticeably only beyond 6m. However, the relative-error-based MRA metric is more tolerant at larger ground-truth distances, while penalizing short-range errors more strictly. Since VSI-Bench distances are skewed toward smaller values, its evaluation is systematically stricter. We argue that our setting better reflects true 3D spatial reasoning, as excessive penalization of short-range errors can obscure model competence on genuinely challenging spatial tasks.

**Do specialized VLMs exhibit stronger 3D reasoning?** We evaluate a range of VLMs explicitly fine-tuned for 3D

*Table 3.* Evaluations on ReVSI (black) and VSI-Bench (light-gray). For ReVSI, each model is evaluated against frame-sampling-specific ground-truth answers corresponding to its inference frames (shown in *Frames* column), whereas VSI-Bench uses a single set of ground-truth answers shared across all frame settings. ReVSI scores that exceed their VSI-Bench counterparts are highlighted in green.

| Method | Frames | Numerical Question | | | | Multiple-Choice Question | | | Avg. |
|---|---|---|---|---|---|---|---|---|---|
| | | Obj. Cnt. | Abs. Dist. | Obj. Size | Room Size | Rel. Dist. | Rel. Dir. | Route Plan | |
| *Baseline* | | | | | | | | | |
| Chance (Random) | ALL | - | - | - | - | 23.7 (25.0) | 26.8 (36.1) | 26.0 (28.3) | - |
| Chance (Frequency) | ALL | 52.2 (62.1) | 40.1 (32.0) | 17.4 (29.9) | 20.9 (33.1) | 25.8 (25.1) | 31.9 (47.9) | 30.2 (28.4) | 31.4 (34.0) |
| *Proprietary Models (API)* | | | | | | | | | |
| GPT-5.2 | 64 | 56.2 (57.1) | 41.5 (33.4) | 73.9 (64.6) | 63.0 (59.0) | 48.4 (48.0) | 34.9 (33.3) | 38.2 (36.7) | 50.9 (49.2) |
| Gemini 3 Flash | 1 FPS | 65.7 (45.6) | 53.1 (36.3) | 77.6 (74.9) | 52.8 (47.4) | 64.6 (54.3) | 47.9 (52.4) | 41.8 (50.0) | 57.6 (55.9) |
| Gemini 3 Pro | 1 FPS | 60.1 (45.3) | 54.7 (38.3) | 79.3 (73.0) | 51.9 (47.4) | 68.1 (70.0) | 56.0 (60.8) | 56.4 (65.3) | 60.9 (60.5) |
| *Open-Source Models* | | | | | | | | | |
| Qwen3-VL-8B-Instruct | 64 | 40.4 (70.0) | 52.3 (50.5) | 69.0 (74.7) | 45.1 (63.3) | 57.1 (57.3) | 39.5 (52.3) | 40.5 (33.5) | 49.1 (57.4) |
| Qwen3-VL-32B-Instruct | 64 | 46.9 (74.0) | 65.0 (57.0) | 70.4 (76.6) | 55.8 (70.8) | 53.8 (55.6) | 34.0 (59.1) | 47.3 (39.7) | 53.3 (61.8) |
| InternVL3.5-8B | 64 | 43.3 (72.7) | 54.6 (40.3) | 64.2 (68.4) | 47.6 (65.3) | 45.0 (57.0) | 36.3 (48.6) | 44.4 (35.6) | 47.9 (55.4) |
| InternVL3.5-38B | 64 | 43.8 (73.9) | 60.6 (39.2) | 70.2 (73.0) | 58.4 (65.0) | 57.4 (66.2) | 45.9 (72.0) | 42.7 (36.1) | 54.1 (60.8) |
| LLaVA-Video-7B-Qwen2 | 64 | 31.3 (50.6) | 1.4 (13.3) | 52.5 (44.7) | 16.7 (23.8) | 38.3 (43.7) | 33.3 (42.7) | 38.4 (35.6) | 30.3 (36.3) |
| LLaVA-Video-72B-Qwen2 | 64 | 40.1 (51.9) | 29.6 (24.0) | 59.3 (57.4) | 27.9 (32.7) | 39.6 (42.4) | 24.8 (37.4) | 43.0 (32.0) | 37.8 (39.7) |

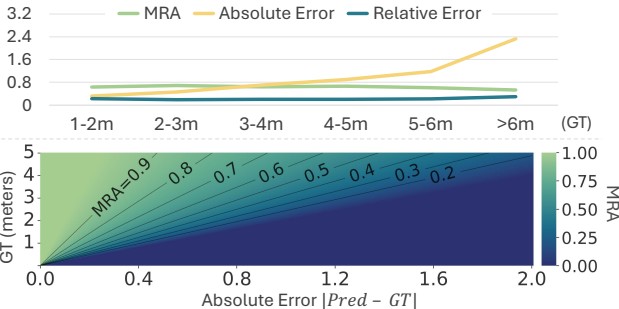

*Figure 7.* Analysis of object absolute distance evaluation. **Up:** Qwen3-VL-32B-Instruct performance across GT distance ranges. Due to the relative-error-based formulation, MRA and relative error remain stable across distances, while absolute error increases with GT distance. **Bottom:** MRA heatmap over GT distance and absolute error, showing stricter penalties at small GT values.

spatial intelligence and compare them against their corresponding base models (Table 4). These specialized models adopt diverse video sampling frames during training and inference. We use our ReVSI's frame-adaptive ground-truth answers and evaluate each model under its native frame setting (*i.e.*, evaluate with the same number of frames the model was trained with).

Contrary to results reported on VSI-Bench, fine-tuned models do not exhibit the large and consistent performance gains suggested by prior evaluations. On ReVSI, all finetuned models show substantially smaller improvements over their base models, and in several cases (*e.g.*, SpaceR (Ouyang et al., 2025)), fine-tuning even leads to performance degradation across multiple tasks. This discrepancy indicates the apparent gains on VSI-Bench do not reliably translate to more robust 3D spatial reasoning.

We attribute this phenomenon to two primary factors. First, most specialized models follow VSI-Bench's data generation pipeline, which relies on noisy 3D annotations and

datasets. As a result, models may learn incorrect answers that interfere with the base model's existing capabilities. Second, as shown in Figure 5, biases in the scene data encourage models to rely on category-level priors or hallucinated cues rather than genuine visual evidence. We further analyze and substantiate this effect in the next subsection.

Finally, we observe that scaling up training data yields marginal performance improvements under the current data construction paradigm. For example, increasing the training size for Spatial-MLLM (Wu et al., 2025a) from 135k to 820k samples results in only marginal gains of ~3%, suggesting that data quality and supervision fidelity, rather than quantity alone, are the primary bottlenecks.

**Hallucination or perception?** We analyze the robustness of 3D reasoning using *dummy-video settings* (see Section 5.2), where queried objects are removed and the ground-truth answer is always 0. As shown in Table 5, human performance exhibits zero hallucination across all dummy-video settings. Proprietary models consistently have lower hallucination rates than open-source models, indicating a stronger tendency to ground predictions in visual evidence rather than priors. In contrast, all specialized finetuned models fail catastrophically on object counting: despite the absence of queried objects, they frequently predict non-zero counts, revealing severe overfitting to noisy training supervision and a systematic disregard of visual input.

Notably, this experiment further exposes a stark behavioral divergence between two strong zero-shot models, InternVL3.5 (Wang et al., 2025) and Qwen3-VL (Qwen Team, 2025). Although both achieve comparable performance on standard object counting tasks, Qwen3-VL correctly predicts 0 when no objects are present and exhibits near-zero hallucination on fully black videos. In contrast, InternVL3.5 is easily misled by dummy videos that retain scene context but lack objects, and consistently predicts 2

*Table 4.* Performance of specialized 3D VLMs and their base models (gray lines) on ReVSI (black numbers) and VSI-Bench (light-gray numbers in parentheses). ReVSI evaluates each model under its native inference frame setting using frame-adaptive ground-truth answers, enabling a fair comparison beyond the fixed-answer evaluation used in VSI-Bench. Scores lower than the base model's are highlighted in red. *Qwen2.5-7B-Instruct+SigLIP2* is left blank, as its module embeddings are not aligned prior to fine-tuning.

| Method | Frames | Numerical Question | | | | Multiple-Choice Question | | | Avg. |
|---|---|---|---|---|---|---|---|---|---|
| | | Obj. Cnt. | Abs. Dist. | Obj. Size | Room Size | Rel. Dist. | Rel. Dir. | Route Plan | |
| *Qwen2.5-7B-Instruct+SigLIP2* | – | – | – | – | – | – | – | – | – |
| Cambrian-S-7B | 128 | 48.4 (73.2) | 60.5 (50.5) | 65.5 (74.9) | 46.7 (72.2) | 37.1 (71.1) | 48.5 (76.2) | 37.0 (41.8) | 49.1 (67.5) |
| *Qwen2.5-VL-7B-Instruct* | 4 FPS | 36.9 (36.8) | 15.0 (17.6) | 49.7 (51.0) | 29.0 (29.2) | 31.5 (35.4) | 29.5 (38.4) | 36.7 (33.5) | 32.6 (32.6) |
| VST-7B-SFT | 4 FPS | 35.4 (72.0) | 52.6 (44.4) | 67.9 (74.3) | 47.2 (68.3) | 49.2 (59.7) | 36.9 (55.8) | 35.4 (44.9) | 46.4 (65.2) |
| *Qwen2.5-VL-7B-Instruct* | 32 | 34.3 (43.7) | 21.7 (22.3) | 45.5 (49.2) | 35.1 (37.5) | 32.6 (40.1) | 33.7 (38.9) | 34.1 (32.0) | 33.9 (37.7) |
| SpaceR-7B (SG-RLVR) | 32 | 30.7 (61.9) | 34.5 (28.6) | 52.0 (60.9) | 18.6 (35.2) | 22.8 (38.2) | 34.5 (46.0) | 20.2 (31.4) | 30.5 (43.5) |
| *Qwen2.5-VL-3B-Instruct* | 16 | 18.7 (24.3) | 15.6 (24.7) | 16.8 (31.7) | – (22.6) | 33.2 (38.3) | 34.3 (41.6) | – (26.3) | 23.7 (30.6) |
| Spatial-MLLM-4B-135k | 16 | 40.7 (65.8) | 45.3 (40.7) | 46.8 (58.3) | – (55.6) | 32.3 (43.2) | 37.4 (55.5) | – (36.1) | 40.5 (50.7) |
| Spatial-MLLM-4B-820k | 16 | 41.5 (66.7) | 40.0 (37.9) | 53.1 (69.7) | – (55.7) | 30.7 (52.0) | 39.2 (54.9) | – (39.7) | 40.9 (53.8) |
| *LLaVA-Video-7B-Qwen2* | 32 | 29.9 (48.5) | 1.5 (14.0) | 53.0 (47.8) | 19.3 (24.2) | 39.1 (43.5) | 33.8 (42.4) | 38.8 (34.0) | 30.8 (36.3) |
| VLM3R-7B | 32 | 41.6 (70.2) | 61.6 (49.4) | 64.8 (69.2) | 52.5 (67.1) | 46.5 (65.4) | 49.5 (80.5) | 34.1 (45.4) | 50.1 (60.9) |

*Table 5.* Object counting results on 16-frame dummy videos where ground-truth answers are always 0. **Query-Drop** removes frames containing the queried object while preserving the surrounding scene and other objects. **First-Frame** repeats the first frame of the Query-Drop video across all 16 frames. **Black** uses a 16-frame black video. Exact Match accuracy is reported over 997 questions, with 0 as the only correct answer.

| Method | Accuracy (Exact Match) | | |
|---|---|---|---|
| | Query-Drop | First-Frame | Black |
| *Zero-shot* | | | |
| Human | 100.0 | 100.0 | 100.0 |
| GPT-5.2 | 74.0 | 89.6 | 99.2 |
| Gemini 3 Pro | 62.3 | 85.0 | 94.0 |
| Qwen2.5-VL-7B-Instruct | 55.8 | 79.9 | 99.8 |
| Qwen3-VL-8B-Instruct | 34.7 | 80.9 | 99.8 |
| Qwen3-VL-32B-Instruct | 50.5 | 92.7 | 100.0 |
| InternVL3.5-8B | 14.7 | 52.5 | 17.7 |
| InternVL3.5-38B | 9.1 | 45.0 | 1.2 |
| LLaVA-Video-7B-Qwen2 | 47.7 | 49.9 | 0.0 |
| LLaVA-Video-72B-Qwen2 | 45.0 | 65.6 | 15.2 |
| *Fine-tuned* | | | |
| Cambrian-S-7B | 1.1 | 2.8 | 0.0 |
| VST-7B-SFT | 1.1 | 8.7 | 0.4 |
| SpaceR-7B (SG-RLVR) | 8.1 | 24.3 | 14.6 |
| Spatial-MLLM-4B-135k | 0.2 | 0.9 | 0.0 |
| Spatial-MLLM-4B-820k | 0.0 | 0.0 | 0.0 |
| VLM3R-7B | 4.1 | 2.5 | 0.2 |

*Table 6.* Object size estimation on 16-frame dummy videos. **Real** uses original videos, while **Black** uses fully black videos. Mean Relative Accuracy (MRA) is reported, higher scores on **Real** and lower scores on **Black** mean less hallucination. The results suggest that InternVL3.5 presents much more severe hallucination than Qwen3-VL.

| Method | Real | Black |
|---|---|---|
| Qwen3-VL-8B-Instruct | 65.3 | 0.7 |
| Qwen3-VL-32B-Instruct | 68.6 | 0.3 |
| InternVL3.5-8B | 65.0 | 50.3 |
| InternVL3.5-38B | 70.2 | 48.6 |

## 7. Conclusion

We revisit the evaluation of 3D spatial intelligence in VLMs and identify fundamental validity issues in existing benchmarks, including uncontrolled frame sampling, ambiguous question construction, and noisy ground-truth annotations. To address these issues, we introduce ReVSI, a rigorously curated benchmark that enforces frame-budget-aware evaluation, visibility-consistent question generation, and systematic human verification across spatial reasoning tasks. Beyond standard accuracy reporting, we propose frame-budgeted evaluation protocols and visibility-guided diagnostics that expose substantial differences in models' sensitivity to visual evidence, reliance on scene priors, and hallucination behavior. These differences are often obscured under conventional evaluation settings. Together, our benchmark and diagnostic tools provide a more reliable foundation for analyzing, comparing, and developing video-language models with robust and grounded 3D spatial reasoning capabilities.

**Limitations & future work.** The high-quality 3D indoor spatial intelligence dataset introduced in this work relies on costly expert-level human annotation, which limits scalability to substantially larger datasets or training-scale supervision. Developing automated or semi-automated pipelines for generating high-quality spatial supervision remains an important direction for future work.

on black videos. This behavior aligns with the dataset bias observed in Figure 5. We hypothesize that this discrepancy arises from data contamination or perceptual limitations, causing InternVL3.5 to memorize scene priors rather than reason from visual evidence.

We validate these findings on object size estimation using an analogous dummy-video protocol (Table 6). Despite explicitly prompting models to answer "based on visual evidence from the video", InternVL3.5 still relies heavily on priors, achieving high task scores even on black videos. In contrast, Qwen3-VL exhibits almost zero hallucination under this setting, reinforcing that strong benchmark performance can mask fundamentally different reasoning behaviors, which are difficult to expose using VSI-Bench alone.

**Acknowledgments.** This work was funded in part by a CIFAR AI Chair and NSERC Discovery Grants, and enabled by support from the Digital Research Alliance of Canada and a CFI/BCKDF JELF grant. We thank Jiayi Liu for help with benchmark data verification and for valuable discussions.

## Impact Statement

Vision-language models are being adopted widely, with users relying on these models to extract information from images and videos, answer questions ranging from identifying products shown in images to soliciting advice based on medical scans. In this work, we aim to provide an improved benchmark for assessing the ability of VLMs to interpret and understand a video of a 3D environment. Accurate assessment of the ability of VLMs to reason about a 3D space is potentially important for applications operating in a 3D environment and requiring 3D understanding and reasoning (*e.g.*, assessment of what is in a room, how far away two objects are from each other). Inaccurate assessment can mislead users in how well these systems actually perform, and cause overconfidence in the ability of VLMs to provide correct answers.

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

# Appendix

Here, we analyze annotation fidelity and frame-budget sensitivity in VSI-Bench (Yang et al., 2025a) (§A), describe ReVSI construction and annotation tools (§B), present dataset statistics and qualitative examples (§C), provide evaluation details (§D), and report additional results (§E).

## A. Additional diagnostics on VSI-Bench

**Annotation fidelity issues.** Examples in Figure 8 show that a non-trivial portion of VSI-Bench (Yang et al., 2025a) questions are affected by annotation inaccuracies due to noisy 3D scene reconstructions. These issues undermine ground-truth reliability and motivate the need for explicit verification and curation when repurposing scanned 3D datasets for video-based spatial evaluation.

**Answerability & correctness vs. frame budget.** Table 7 shows that question validity in VSI-Bench (Yang et al., 2025a) degrades substantially as the frame budget decreases. Many questions become unanswerable or incorrect under commonly used sampling settings.

*Table 7.* Breakdown of answerability and correctness of VSI-Bench questions under different frame-sampling settings. A dash in the Correct column indicates tasks where correctness trivially matches answerability, as the ground-truth answer remains valid whenever the required objects appear in the sampled frames.

| Frame | Task Type | Correct | Answerable | Total |
|---|---|---|---|---|
| 16 | Obj. Cnt. | 432 (77%) | 535 (95%) | 565 |
| | Abs. Dist. | – | 644 (77%) | 834 |
| | Obj. Size | – | 824 (86%) | 953 |
| | Rel. Dist. | 384 (54%) | 392 (55%) | 710 |
| | Rel. Dir. | – | 676 (70%) | 968 |
| | Appr. Order | 174 (28%) | 230 (37%) | 618 |
| 32 | Obj. Cnt. | 500 (88%) | 555 (98%) | 565 |
| | Abs. Dist. | – | 766 (92%) | 834 |
| | Obj. Size | – | 915 (96%) | 953 |
| | Rel. Dist. | 565 (80%) | 570 (80%) | 710 |
| | Rel. Dir. | – | 872 (90%) | 968 |
| | Appr. Order | 296 (48%) | 391 (63%) | 618 |
| 64 | Obj. Cnt. | 533 (94%) | 561 (99%) | 565 |
| | Abs. Dist. | – | 805 (97%) | 834 |
| | Obj. Size | – | 937 (98%) | 953 |
| | Rel. Dist. | 650 (92%) | 652 (92%) | 710 |
| | Rel. Dir. | – | 932 (96%) | 968 |
| | Appr. Order | 431 (70%) | 486 (79%) | 618 |

## B. ReVSI construction details

### B.1. Annotation tools

We develop a suite of web-based tools covering object visibility annotation, 3D bounding box refinement, room boundary annotation, and end-to-end data verification (see Figures 9 to 12). All annotations are performed directly on video-aligned 3D reconstructions, enabling annotators to cross-check raw videos, rendered geometry, and sampled frames. All annotations are manually performed by the authors with domain expertise in 3D vision, and ReVSI contains no model-generated annotations.

### B.2. GPT verification for object naming

We use GPT-5.2 (OpenAI, 2025) as an auxiliary tool to verify object names only for cases where annotators are uncertain about the object name during 3D object annotation. For each object, annotators manually select 1–3 representative video frames in which the object is clearly visible with minimal occlusion, and crop tight image regions that primarily contain the object with limited background context. These cropped images are then provided to GPT-5.2 with a simple prompt ("These are images of an object. What is the name of the object?") to obtain an independent prediction.

Annotators compare the GPT-generated name with their own label as a consistency check. The final object label is always determined by the human annotator. If a clear agreement cannot be reached, the object is discarded to avoid introducing ambiguity. Since this verification step is integrated into the per-object annotation workflow, we use the ChatGPT web interface for convenience.

### B.3. Gravity-aligned 3D bounding box

We initialize all 3D boxes using a gravity-aligned oriented bounding box (OBB) algorithm (Algorithm 1), followed by manual refinement to ensure accurate and compact localization. Figure 13 compares our approach with the OBB construction used in VSI-Bench (Yang et al., 2025a).

---

**Algorithm 1** Gravity-Aligned Oriented Bounding Box

**Require:** Points $P = \{p_i\}_{i=1}^N$, $p_i \in \mathbb{R}^3$, up-axis $\vec{z} = [0, 0, 1]^T$
**Ensure:** OBB $(C, R, S)$: center $C \in \mathbb{R}^3$, rotation $R \in \mathbb{R}^{3 \times 3}$, side lengths $S \in \mathbb{R}^3$
1: $q_i \leftarrow (p_{i,x}, p_{i,y}) \in \mathbb{R}^2$ {$xy$-projection}
2: $V \leftarrow \text{Hull}(\{q_i\})$ {ordered 2D convex-hull vertices}
3: $A_{\min} \leftarrow +\infty$
4: **for each** hull edge direction $\vec{v}$ from consecutive vertices in $V$ **do**
5:    $\vec{v} \leftarrow \vec{v}/\|\vec{v}\|, \quad \vec{w} \leftarrow [-v_y, \ v_x]^T$
6:    $W \leftarrow \max_{q \in V} \vec{v}^T q - \min_{q \in V} \vec{v}^T q$
7:    $D \leftarrow \max_{q \in V} \vec{w}^T q - \min_{q \in V} \vec{w}^T q$
8:    **if** $WD < A_{\min}$ **then**
9:       $A_{\min} \leftarrow WD, \quad \vec{x}_{\text{best}} \leftarrow (W \geq D) ? \vec{v} : \vec{w}$
10:    **end if**
11: **end for**
12: $\vec{x} \leftarrow \text{norm}([x_{\text{best},x}, x_{\text{best},y}, 0]^T), \quad \vec{y} \leftarrow \text{norm}(\vec{z} \times \vec{x})$
13: $R \leftarrow [\vec{x} \ \vec{y} \ \vec{z}]$ {columns are local axes (local→world)}
14: $p_i^\ell \leftarrow R^T p_i \ \forall i$
15: $\mathbf{m} \leftarrow \min_i p_i^\ell, \quad \mathbf{M} \leftarrow \max_i p_i^\ell$ {elementwise}
16: $S \leftarrow \mathbf{M} - \mathbf{m}, \quad C_\ell \leftarrow \frac{1}{2}(\mathbf{m} + \mathbf{M}), \quad C \leftarrow RC_\ell$
17: **return** $(C, R, S)$

---

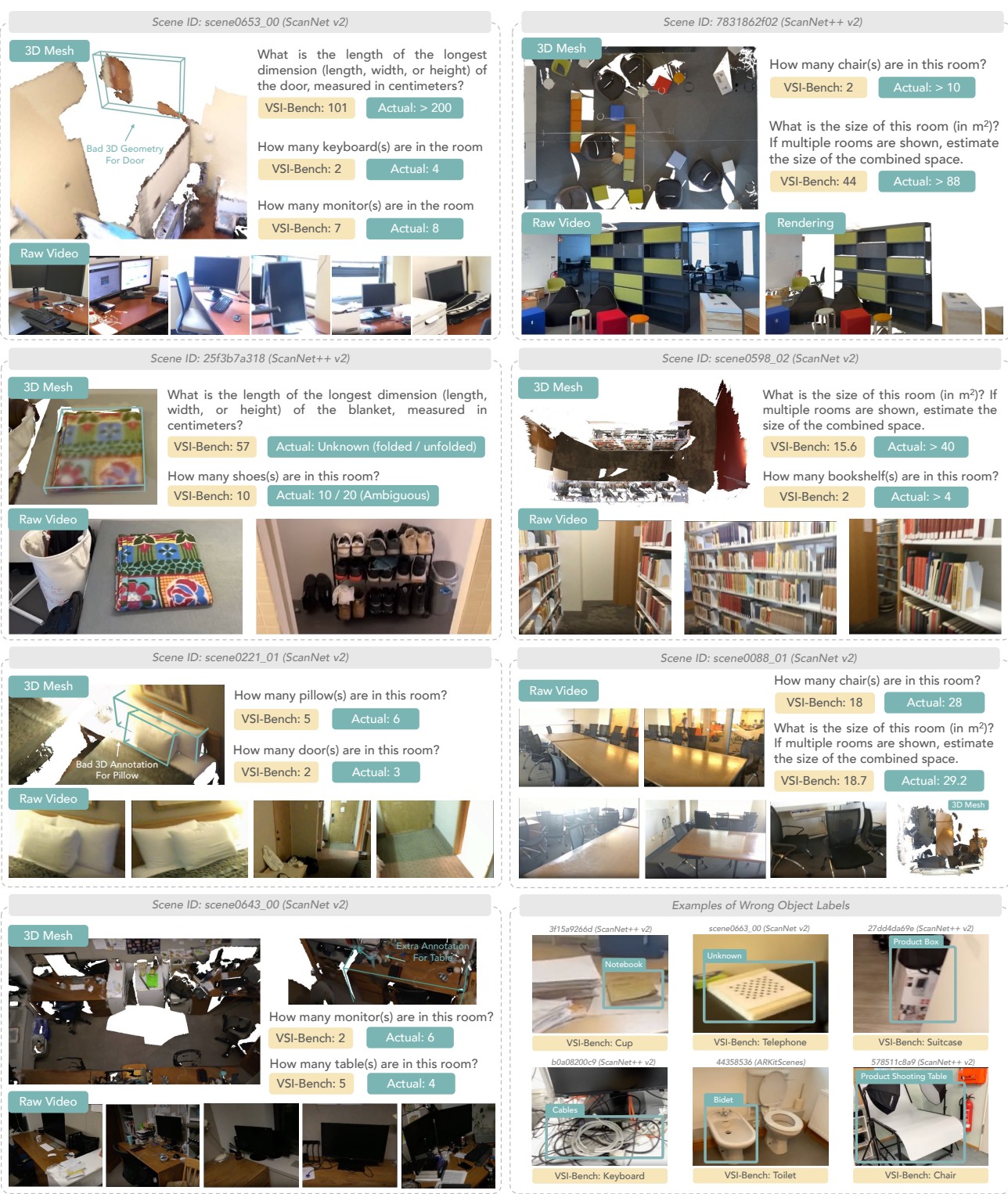

*Figure 8.* Example of ground-truth errors in VSI-Bench (Yang et al., 2025a). Common issues include incorrect object annotation and broken geometry, stemming from noisy 3D scene reconstruction and annotations in the underlying datasets (Dai et al., 2017; Yeshwanth et al., 2023; Baruch et al., 2021; Wald et al., 2019; Mao et al., 2022). The absence of systematic manual verification allows these errors to propagate into the VSI-Bench. Note that some questions don't have a precise actual answer as they are not included in ReVSI.

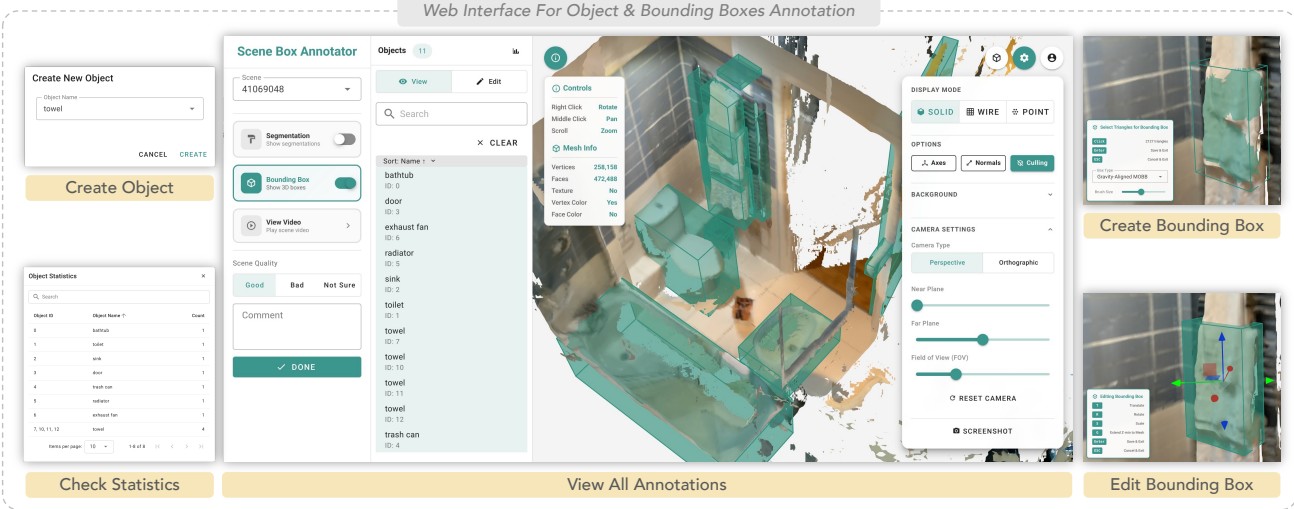

*Figure 9.* ReVSI object box annotation interface. Our web interface allows annotators to view and check the bounding boxes of all objects. In some cases, objects may be missed in the original 3D scene annotations or have inaccurate bounding boxes. Users can then create a new object, select triangles to define (or correct) the object segmentation to obtain an initial bounding box (algorithmically computed). The user can then edit and refine the bounding box.

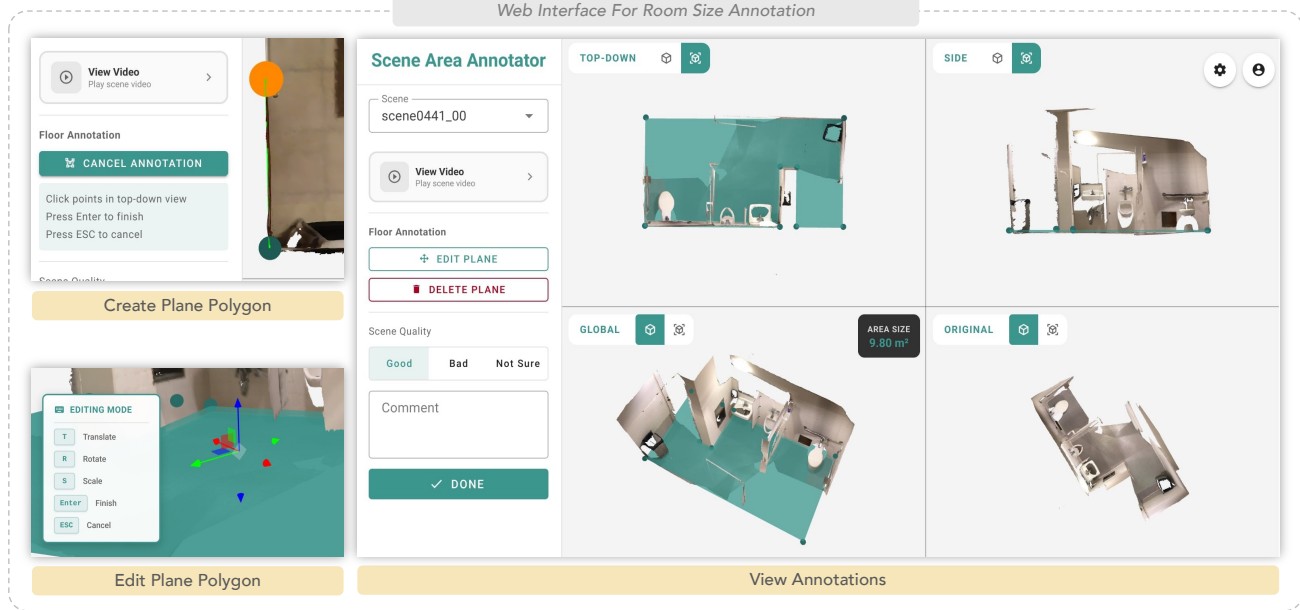

*Figure 10.* ReVSI room size annotation interface. Due to noisy 3D reconstructions, automatic methods for determining the room layout and size can be inaccurate. We ask annotators to provide the floor boundary by clicking corner points on the floor plane. The interface allows annotators to check the accuracy of the floor mask from different views. The room size is then computed based on the annotated polygon of the floor area.

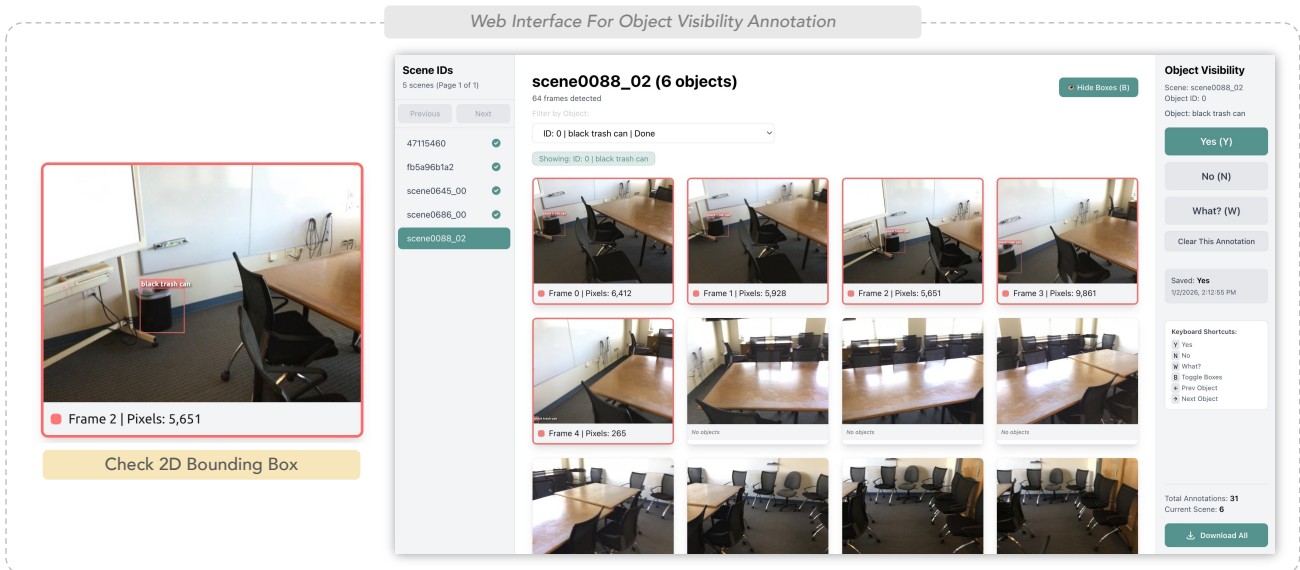

*Figure 11.* ReVSI object visibility annotation interface. The web tool enables annotators to label object visibility under different frame-sampling settings (16/32/64/All). For each object, the interface visualizes its 2D projected bounding box and pixel coverage across frames, supports click-based zoom-in inspection, and allows efficient visibility annotation at the object level.

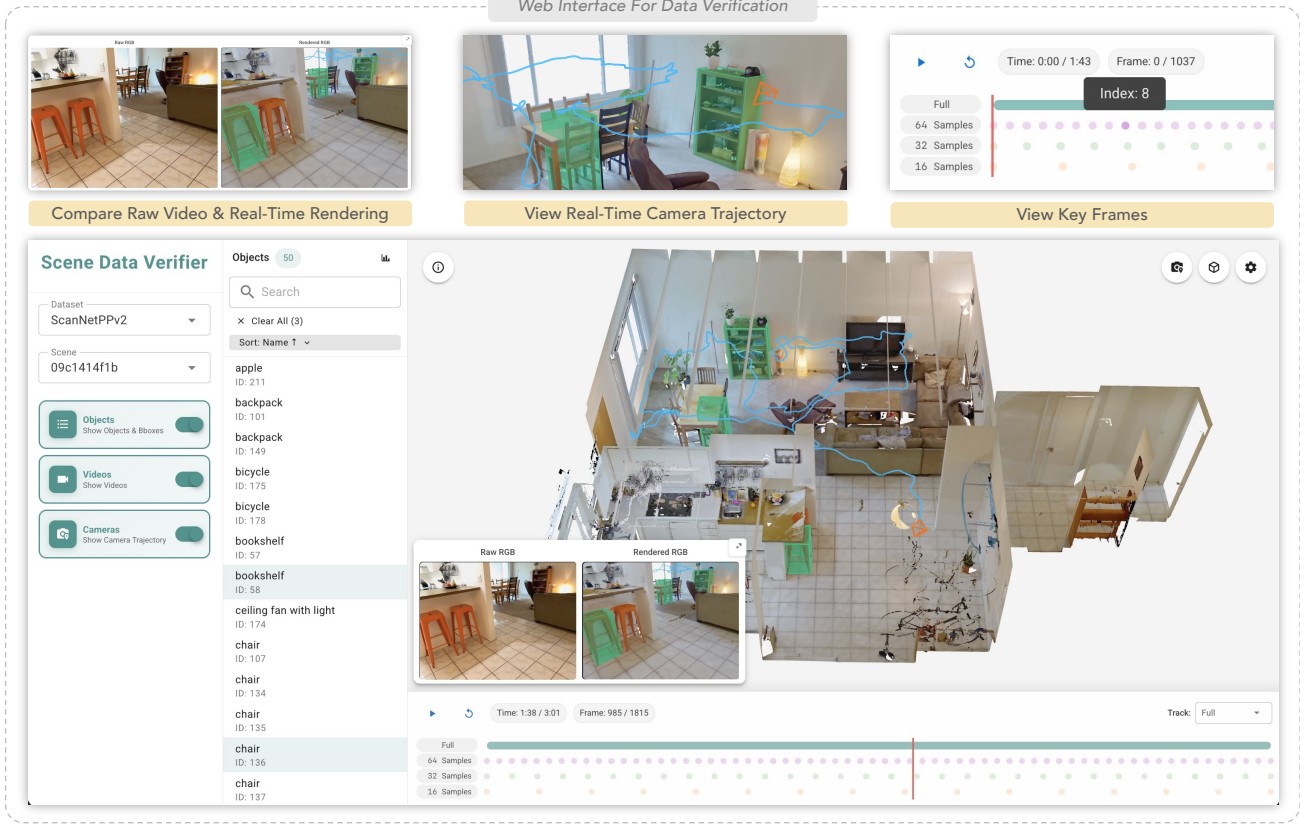

*Figure 12.* ReVSI data verification interface. To ensure that the dataset can be used to generate accurate answers, we implement an interface for inspecting and verifying the scene data. Our interface shows the 3D scene (bottom), with a panel for users to select and inspect the labeled objects. We also allow users to compare the raw video and real-time rendering (top left), visualize the camera trajectory (top middle), and check the sampled frames (top right).

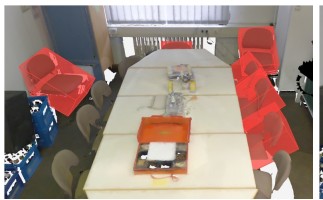 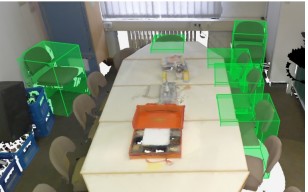

VSI-Bench (OBB)          ReVSI (Gravity-Aligned OBB)

*Figure 13.* Comparison of 3D object bounding boxes used in VSI-Bench (Yang et al., 2025a) and ReVSI. Both are computed from 3D segmentation masks. VSI-Bench relies solely on Open3D (Zhou et al., 2018) minimum oriented bounding box (OBB) algorithm. However, in real-world 3D scans, segmentation mask noise can lead to inaccurate convex hull estimation, resulting in erroneous box rotation and tilt. In contrast, our bounding boxes are initialized using a gravity-aligned OBB algorithm (Algorithm 1) and subsequently manually refined to ensure accurate and compact localization.

### B.4. 2D bounding box guidance

To accelerate and improve the quality of object visibility annotation across video frames, we precompute per-frame 2D bounding boxes for each object in our annotation tool (Figure 11) as auxiliary guidance.

Specifically, based on the ReVSI 3D object bounding box annotations and meshes, we approximate a pseudo 3D segmentation mask by treating all mesh triangles within each bounding box as belonging to the object. Using the camera trajectories provided by the scene datasets, we perform ray casting from each viewpoint to obtain per-frame object pixels, from which 2D bounding boxes are derived. See Figure 14 for the full process.

Due to inaccuracies in camera poses and noise in reconstructed meshes from the scene datasets, these projected 2D bounding boxes may exhibit misalignment and are therefore not reliable for directly determining object visibility. Instead, they are used solely as auxiliary cues to assist annotation. All final annotations are manually done by annotators to ensure high quality.

### B.5. Video sampling details

**All-frame.** For ScanNet v2 (Dai et al., 2017) and ScanNet++ v2 (Yeshwanth et al., 2023), videos are downsized to $640 \times 480$ and sampled at 10 fps. Frames with invalid camera poses (i.e., containing `NaN` values) are discarded for ScanNet v2. For ARKitScenes (Baruch et al., 2021) and MultiScan (Mao et al., 2022), videos are first rotated to enforce a sky-up orientation, and then downsized to $640 \times 480$ or $480 \times 640$ at 10 fps based on the video orientation. Due to frequent errors in orientation metadata, particularly in ARKitScenes, all videos are manually verified to ensure correct alignment. For 3RScan (Wald et al., 2019), videos are resized to $360 \times 640$ and sampled at 4 fps, due to the sparsity of the officially provided frames.

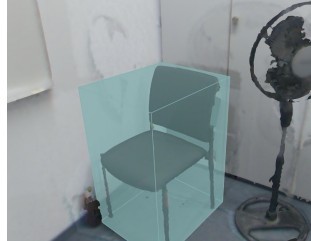 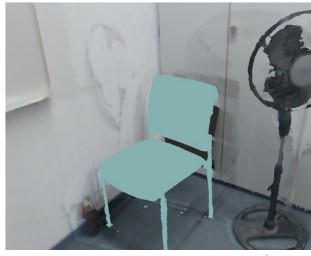

1. 3D Bounding Box          2. 3D Segmentation (Pseudo)

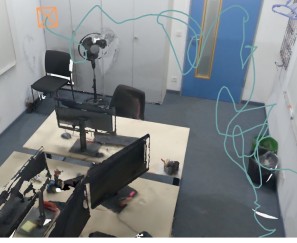 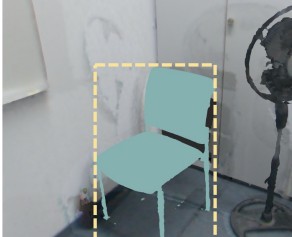

3. Ray Casting From Views          4. 2D Bounding Box

*Figure 14.* Pipeline for computing auxiliary 2D bounding boxes for object visibility annotation guidance. We start with ReVSI 3D object bounding box annotations (1), and use the meshes with the bounding boxes to approximate pseudo 3D object masks (2). We then project the masks into each view via ray casting using camera trajectories provided by scene datasets (3), and derive per-frame 2D masks and bounding boxes (4).

**16/32/64-frame.** We construct fixed frame-budget subsets via hierarchical uniform sampling. Starting from all-frame, we first apply `np.linspace` (Harris et al., 2020) to obtain the 64-frame. We then recursively apply `np.linspace` on the sampled frames to obtain 32- and 16-frame subsets. This ensures a nested structure: the 16-frame set is a subset of the 32-frame set, which is a subset of the 64-frame set. All subsets for a given video span the same temporal duration, ensuring that corresponding frames share consistent timestamps across different frame budgets. For reproducibility, we provide the sampled frame indices for each video and each frame budget.

**Dummy videos.** We construct three types of dummy videos (16 frame): **Query-Dropped** where we only sample frames that do not contain the queried object, **First-Frame Repeated** where the first frame of the query-dropped video is repeated for all frames, and **Black** where each frame is a dummy black frame. Examples of the three types of dummy videos are shown in Figure 15.

### B.6. Prompt & QA templates

We largely follow the prompt and question template design of VSI-Bench (Yang et al., 2025a) to ensure comparability, while introducing targeted refinements to improve robustness, as shown in Table 8. In addition, Table 9 shows that we retain identical templates where applicable while introducing additional variants to increase task difficulties without altering task semantics.

*Table 8.* Evaluation prompt templates used in VSI-Bench (Yang et al., 2025a) and ReVSI. We follow VSI-Bench's protocol that formats prompts as: [VIDEO FRAMES][PRE-PROMPT][QUESTION][POST-PROMPT]. Differences are highlighted using colors. For numerical questions, we explicitly require answers to be provided in Arabic numerals (e.g., "3", "1.8"), as VSI-Bench prompts can elicit word-form responses (e.g., "three"), which are not reliably parsed by their evaluator and lead to evaluation errors.

| Type | Benchmark | Question | Prompt Template |
|---|---|---|---|
| Pre | – | – | *These are frames of a video.* |
| Post | VSI-Bench (Proprietary Models) | Numerical Question Multiple-Choice Question | Do not respond with anything other than a single number! *Answer with the option's letter from the given choices directly.* |
| | VSI-Bench (Open-source Models) | Numerical Question Multiple-Choice Question | *Please answer the question using a single word or phrase.* *Answer with the option's letter from the given choices directly.* |
| | ReVSI (Proprietary Models) | Numerical Question Multiple-Choice Question | Do not respond with anything other than a single number! *Answer with the option's letter from the given choices directly.* |
| | ReVSI (Open-source Models) | Numerical Question Multiple-Choice Question | *Answer the question using a single integer or decimal number.* *Answer with the option's letter from the given choices directly.* |

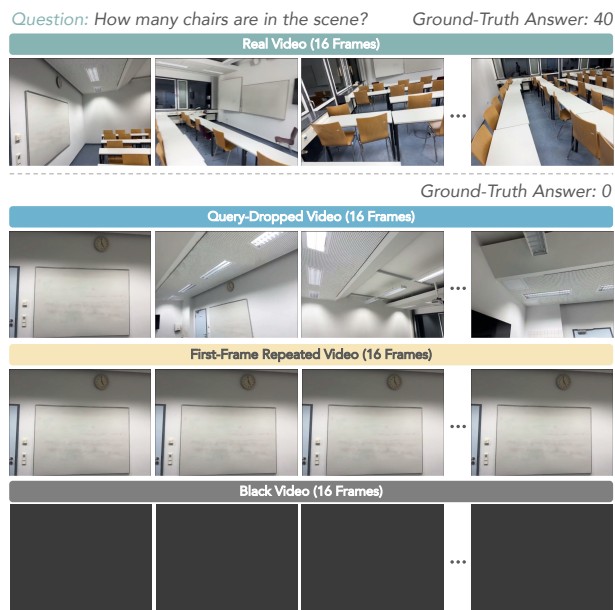

*Figure 15.* Illustration of three dummy video constructions used to probe reliance on visual evidence under the 16-frames setting in ReVSI. **Query-Dropped** video uniformly samples frames that exclude the queried object. **First-Frame Repeated** video repeats the first frame of the query-dropped video across all frames. **Black** video contains only black frames. All dummy videos are assigned a ground-truth count of 0 for object counting questions.

## B.7. Object selection and filtering

We apply task-specific object selection rules during benchmark construction to improve evaluation reliability. Table 10 lists object categories excluded from individual tasks to avoid ill-posed or ambiguous queries and Table 11 summarizes the subsampling strategy used for object size estimation.

## B.8. QA construction

We follow the data generation pipeline of VSI-Bench, while introducing targeted refinements to improve question quality and reduce ambiguity for several tasks.

**Object absolute distance.** We remove questions with object distances below 1 m to avoid trivial cases. To promote a more balanced answer distribution, we further subsample questions with object distances in the ranges of 1–2 m and 2–3 m at rates of 80% and 50%, respectively.

**Object relative direction.** To reduce ambiguity in localizing object positions, we exclude questions involving objects whose extent along either the $x$ or $y$ axis (parallel to the floor plane) exceeds 1.4 m. We additionally filter out questions where any inter-object distance is below 1.5 m, or where the relative angle of any object pair lies within 35° of boundary directions (0°, 90°, 135°, 180°, 225°, 270°), as these cases are prone to corner-case ambiguity.

**Object relative distance.** We remove questions where the relative distance difference between any query object and the anchor object is less than 0.3 m, which may lead to ambiguous comparisons. We also exclude cases where the distance between query and anchor objects is below 1 m to avoid trivial instances.

**Route planning.** We strictly follow the question templates from VSI-Bench and manually re-annotate all questions in this task. Our questions are newly constructed and do not overlap with those in VSI-Bench. During navigation, we incorporate auxiliary anchors beyond annotated objects, including architectural elements such as walls.

*Table 9.* Question templates used in VSI-Bench ([Yang et al., 2025a](#)) and ReVSI. Differences are highlighted using colors. While following the same template style as VSI-Bench, ReVSI introduces additional variants for each question type.

| | VSI-Bench | ReVSI |
|---|---|---|
| **Object Counting** | *How many `obj(s)` are in this room?* | 1. *How many `obj(s)` are in the scene?*
2. *How many `obj_1(s)` and `obj_2(s)` are in the scene in total?* |
| **Absolute Distance** | *Measuring from the closest point of each object, what is the direct distance between the `obj_1` and the `obj_2` (in meters)?* | |
| **Object Size Estimation** | *What is the length of the longest dimension (length, width, or height) of the `obj`, measured in centimeters?* | *Based on visual evidence from the video, what is the length of the longest dimension (length, width, or height) of the `obj`, measured in centimeters?* |
| **Room Size Estimation** | *What is the size of this room (in square meters)? If multiple rooms are shown, estimate the size of the combined space.* | 1. *What is the size of the scene (in square meters)? If multiple rooms are shown, estimate the size of the combined space.*
2. *What is the size of the main room (in square meters)? If multiple rooms are shown, estimate only the size of the dominant room in which the video is primarily recorded.* |
| **Relative Distance** | *Measuring from the closest point of each object, which of these objects (`obj_1`, `obj_2`, `obj_3`, `obj_4`) is the closest to the `obj_5`?* | 1. *Measuring from the closest point of each object, which of these objects (`obj_1`, `obj_2`, `obj_3`, `obj_4`) is the closest to the `obj_5`?*
2. *Measuring from the closest point of each object, which of these objects (`obj_1`, `obj_2`, `obj_3`, `obj_4`) is the farthest from the `obj_5`?* |
| **Relative Direction** | 1. *If I am standing by the `obj_1` and facing the `obj_2`, is the `obj_3` to the left or the right of the `obj_2`?*
2. *If I am standing by the `obj_1` and facing the `obj_2`, is the `obj_3` to my left, right, or back? An object is to my back if I would have to turn at least 135 degrees in order to face it.*
3. *If I am standing by the `obj_1` and facing the `obj_2`, is the `obj_3` to my front-left, front-right, back-left, or back-right? Directions refer to the quadrants of a Cartesian plane (assuming I am at the origin and facing the positive y-axis).* | 1. *If I am standing by the `obj_1` and facing the `obj_2`, is the `obj_3` to my left, right, or back? An object is to my back if I would have to turn at least 135 degrees in order to face it.*
2. *If I am standing by the `obj_1` and facing in the opposite direction of the `obj_2`, is the `obj_3` to my left, right, or back? An object is to my back if I would have to turn at least 135 degrees in order to face it.*
3. *If I am standing by the `obj_1` and facing the `obj_2`, is the `obj_3` to my front-left, front-right, back-left, or back-right?*
4. *If I am standing by the `obj_1` and facing in the opposite direction of the `obj_2`, is the `obj_3` to my front-left, front-right, back-left, or back-right?* |
| **Route Planning** | *You are a robot beginning at the `obj_1` and facing the `obj_2`. You want to navigate to `obj_3`. You will perform the following actions (Note: for each [please fill in], choose either 'turn back,' 'turn left,' or 'turn right.'): 1. Go forward until the `obj_x` 2. [please fill in] 3. Go forward until the `obj_y` 4. [please fill in] 5. Go forward until the `obj_z`.... You have reached the final destination.* | |

*Table 10.* Object names excluded for each question type to reduce annotation ambiguity (e.g., inconsistent singular or plural interpretations), eliminate trivial or ill-defined cases (e.g., objects with limited size variation within each category), and avoid non-rigid objects that do not admit stable geometric definitions. For distance-based tasks, we further exclude large irregular objects with overestimated 3D bounding boxes and ceiling-mounted objects to avoid trivial spatial priors.

| Question Type | Excluded Object Names |
| --- | --- |
| Object Counting | bi-fold door, blinds, clothes, light switch, roller blind, shoes, slippers, window |
| Absolute Distance | L-shape bench, L-shape couch, L-shape sofa, bed, shower seat, u-shape couch |
| Relative Distance | L-shape bench, L-shape couch, L-shape sofa, bathroom ceiling heater, bed, ceiling fan, ceiling fan with light, ceiling lamp, ceiling light, recessed downlight, shower seat, u-shape couch |
| Object Size Estimation | adult bed, apron, backpack, bed, bed frame, bidet, bi-fold door, blinds, bunk beds, ceiling light, ceiling-mounted projector, clothes, coffee cup, comforter, computer mouse, computer tower, cup, curtain, desk phone, dishwasher, dog, door, dryer, dryer sheets box, electric kettle, exhaust fan, fireplace, football, hair dryer, headphones, intercom, iron, jacket, keyboard, kitchen apron, laptop, light switch, mattress, microwave, mirror, mirror, monitor, mug, paper towel, paper towel roll, potted plant, power strip, projector, projector remote, range hood, range oven, recessed downlight, remote controller, roller blind, room door, room entrance door, scissor, shoes, shower curtain, shower seat, shower towel, sink, slippers, smartphone, smoke detector, starbucks cup, steam iron, table phone, telephone, thermostat, tissue box, toilet, toilet brush, toilet paper, toilet paper roll, tv remote, tv remote control, wall clock, wall light, washer, washing machine, water filter pitcher, window, wooden door, wooden room door, yoga mat, yoga mat roll |

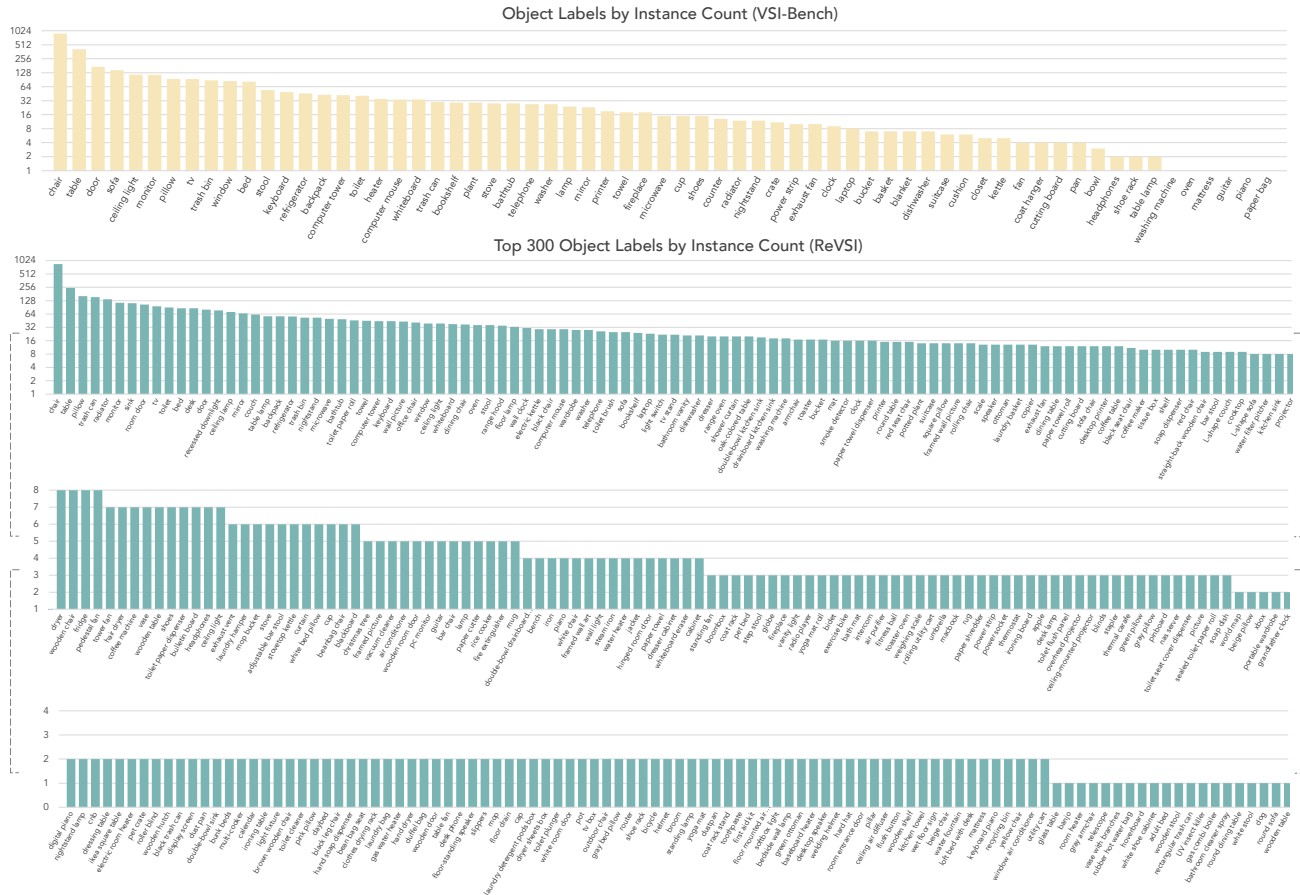

*Figure 16.* Comparison of object label distributions between VSI-Bench ([Yang et al., 2025a](#)) and ReVSI, ranked by instance count. VSI-Bench uses a closed set of 65 object labels, while ReVSI adopts an open-vocabulary setting. We show the top 300 object labels for ReVSI (out of 504 total) for clarity. ReVSI exhibits a longer-tailed distribution with greater label diversity.

*Table 11.* Object sampling configuration for object size estimation in ReVSI. To prevent the model from relying on semantic priors (guessing based on typical sizes) rather than visual evidence, we subsample objects with dominant sizes. The table lists the valid size range constraints (lower – upper bound) in centimeters and sampling rates for each category. "ALL" indicates the object is sampled regardless of size.

| Object Name | Size Range (CM) | Sample Rate |
|---|---|---|
| armchair | 75 – 95 | 5% |
| bathtub | 150 – 170 | 0.5% |
| chair | 75 – 95 | 5% |
| clock | 20 – 40 | 40% |
| double-bowl kitchen sink | 70 – 90 | 10% |
| mirror | ALL | 40% |
| office chair | 75 – 95 | 5% |
| oven | ALL | 40% |
| radiator | 70 – 100 | 1% |
| refrigerator | 170 – 200 | 5% |
| sofa chair | 75 – 95 | 5% |
| table | ALL | 40% |
| trash bin | 20 – 40 | 5% |
| trash can | 20 – 40 | 5% |
| tv | ALL | 20% |
| wardrobe | 180 – 200 | 10% |

## C. ReVSI data statistics & examples

### C.1. Object label statistics

Figure 16 highlights that ReVSI features a substantially broader and more diverse object vocabulary than VSI-Bench (Yang et al., 2025a). The resulting long-tailed label distribution reflects greater coverage of real-world objects and reduces reliance on a small set of frequent categories, encouraging evaluation beyond category-level priors.

### C.2. Object annotation examples

Figure 17 compares the 3D bounding box annotations between VSI-Bench (Yang et al., 2025a) and ReVSI. ReVSI provides more accurate and complete object annotations than VSI-Bench, with tighter bounding boxes, corrected orientations, and improved coverage of previously missing objects. These refinements make ReVSI more representative of real-world indoor scenes and better suited for evaluating spatial reasoning beyond category-level priors. Crucially, high-quality box annotations are foundational to our benchmark, as all spatial questions such as object size estimation and inter-object absolute distance, are constructed using object bounding boxes as geometric proxies.

### C.3. Room area annotation examples

Figure 18 compares the automatically computed room mask used by VSI-Bench (Yang et al., 2025a) vs our manually annotated room polygons from ReVSI. The difference in the room mask illustrates that ReVSI provides more ac-

curate room area definitions and measurements than VSI-Bench.

## D. ReVSI evaluation details

For zero-shot models, we adopt ModelScope SWIFT (Zhao et al., 2024) and LMMs-Eval (Zhang et al., 2024) as inference and evaluation frameworks, using greedy decoding to ensure reproducibility. For fine-tuned models, we use their provided evaluation code and default settings. For Spatial-MLLM (Wu et al., 2025a), we use the non-SA sampling version for simplicity.

For proprietary models, to reduce evaluation cost, we conduct experiments on a *tiny* subset of 1,093 samples from ReVSI (approximately 16% of the full dataset), and use the official tiny set for VSI-Bench (Yang et al., 2025a). For Gemini 3 models (Google DeepMind, 2025), we follow VSI-Bench and use FPS-based video sampling instead of fixed frame counts, consistent with Gemini's default video sampling strategy.

## E. Additional experiments

**Results under different frame-samplings.** We report additional evaluation results (Tables 12 and 13) on ReVSI under 32 and 16 sampled frames to complement the main experiments, which were at 64 sampled frames (Table 3). They further illustrate how model performance varies with available visual evidence.

**Fine-grained performance analysis.** We also present a fine-grained analysis in Table 14 by decomposing each task into its variants: counting when considering single vs multiple objects, estimating room size for single vs multiple rooms, determining which objects are closest or farthest apart (relative distance), and determining the relative direction when considering objects in front or toward the back. Our results reveal systematic performance asymmetries across different question formulations. For instance, models are able to achieve higher accuracies for counting single objects than when there are multiple objects involved. It is also easier for the models to assess which object is the closest than the farthest, and to handle forward (vs backward) facing directional queries.

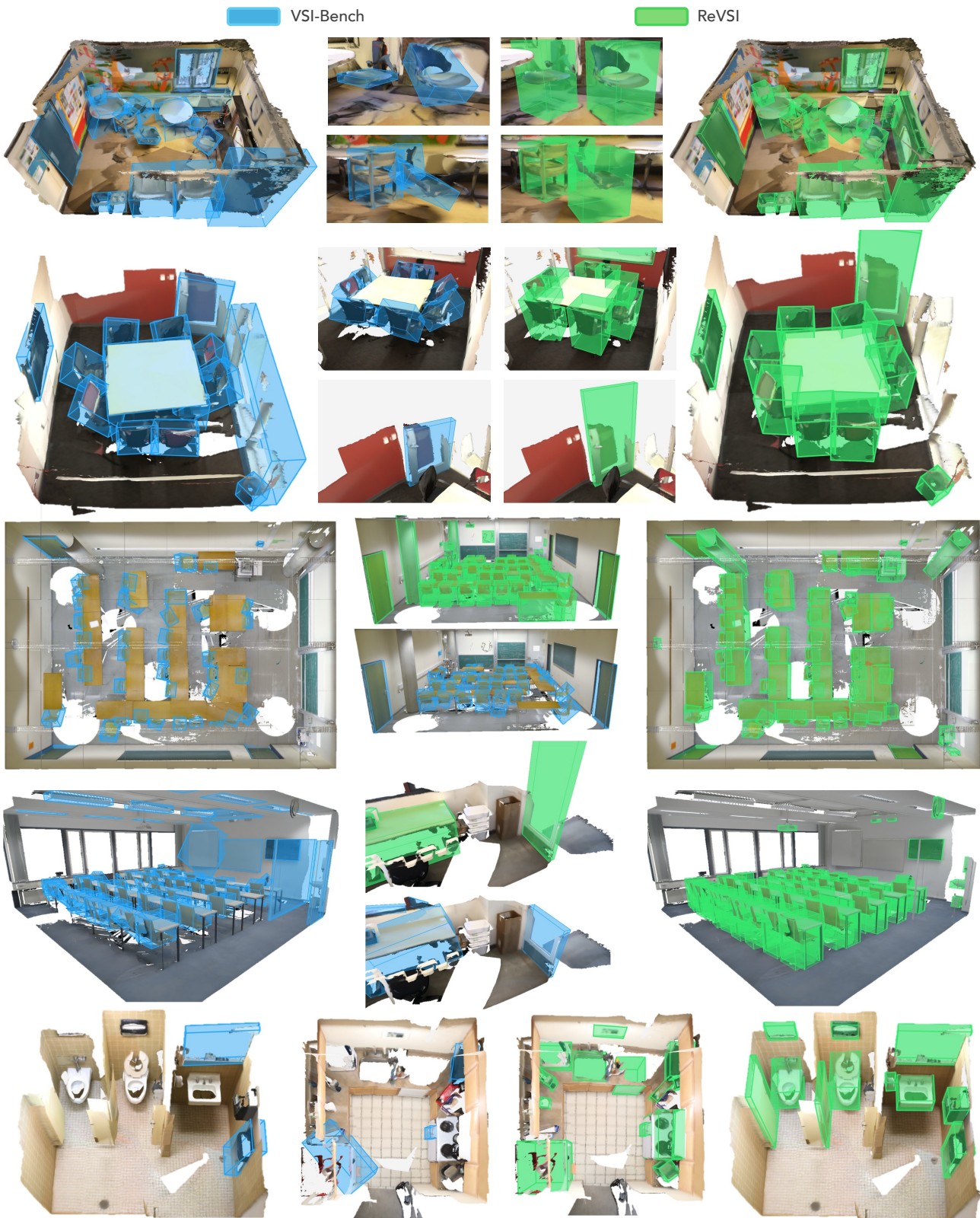

*Figure 17.* Comparison of 3D object bounding box annotations in VSI-Bench (Yang et al., 2025a) (blue) and ReVSI (green). ReVSI provides tighter and more complete boxes with more accurate scale and orientation, correcting missing geometry and misaligned extents in prior annotations. In addition, ReVSI substantially expands object coverage with diverse, open-vocabulary categories, better reflecting real-world indoor scenes and enabling more realistic spatial reasoning evaluation. The richer annotations also serve as reusable assets for downstream scene perception tasks beyond benchmark evaluation.

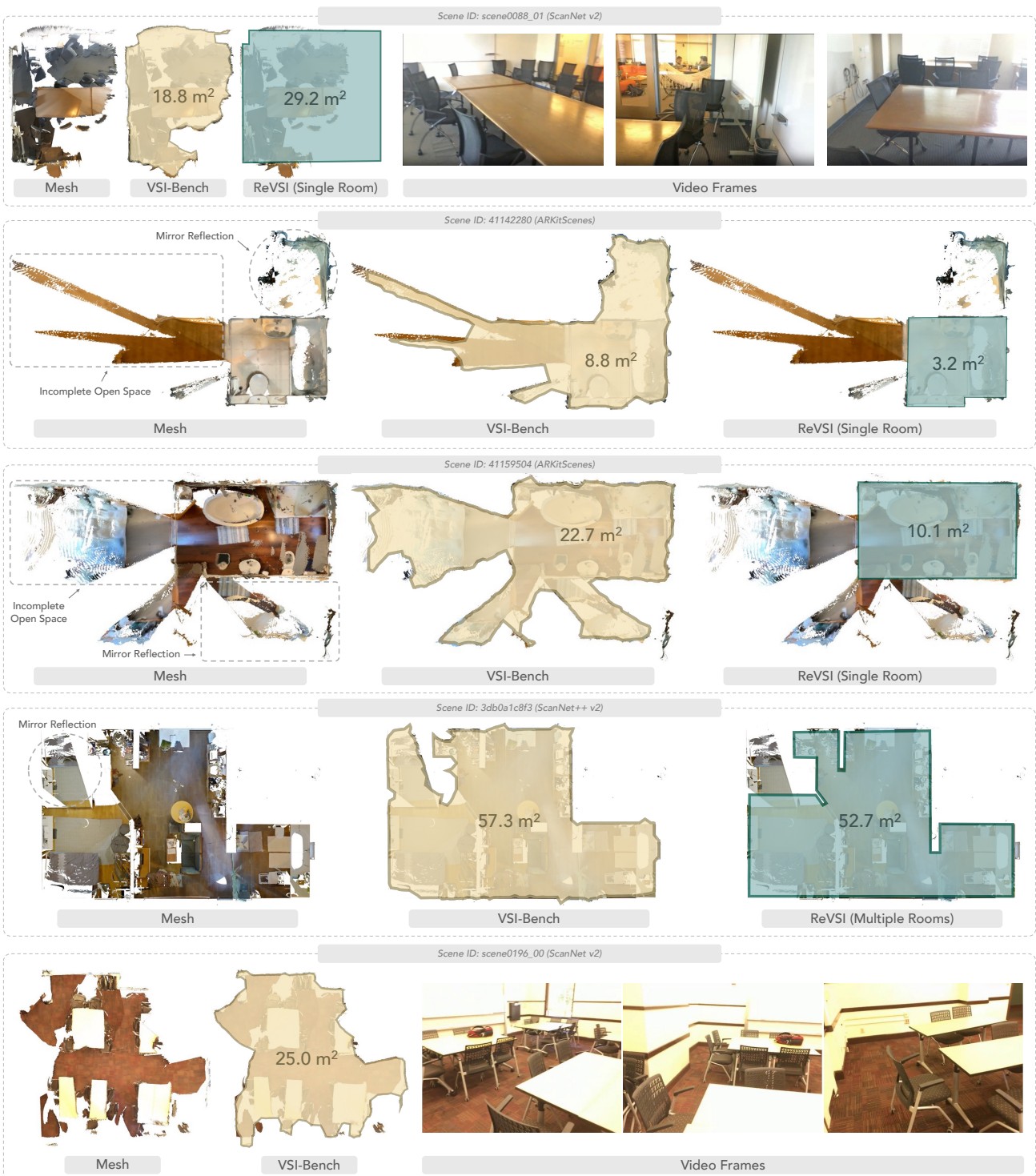

*Figure 18.* Comparison of room area calculation in VSI-Bench ([Yang et al., 2025a](#)) and ReVSI. VSI-Bench computes the room area from the Alpha Shape ([Edelsbrunner et al., 1983](#)) algorithm on noisy reconstructed 3D geometry, while ReVSI room polygons are manually annotated by referring to both 3D and the raw video. Moreover, VSI-Bench always queries the total area of all rooms, which is ambiguous in open-plan scenes, whereas ReVSI introduces flexible templates to distinguish between the area of the main room and that of all visible rooms. Note that scenes with severely incomplete geometry are excluded in ReVSI.

*Table 12.* Evaluation results on ReVSI under the 32-frame sampling setting.

| Method | Numerical Question | | | | Multiple-Choice Question | | | Avg. |
|---|---|---|---|---|---|---|---|---|
| | Obj. Cnt. | Abs. Dist. | Obj. Size | Room Size | Rel. Dist. | Rel. Dir. | Route Plan | |
| *Baseline* | | | | | | | | |
| Chance (Random) | - | - | - | - | 24.6 | 29.3 | 27.9 | - |
| Chance (Frequency) | 51.1 | 40.3 | 17.1 | 20.9 | 26.2 | 32.7 | 30.2 | 31.2 |
| *Open-Source Models* | | | | | | | | |
| Qwen3-VL-8B-Instruct | 36.2 | 47.1 | 67.2 | 45.3 | 50.2 | 37.5 | 36.0 | 45.6 |
| Qwen3-VL-32B-Instruct | 42.2 | 58.3 | 68.5 | 58.6 | 47.9 | 36.6 | 46.5 | 51.2 |
| InternVL3.5-8B | 43.8 | 49.4 | 64.3 | 44.8 | 45.3 | 37.9 | 44.6 | 47.2 |
| InternVL3.5-38B | 43.4 | 56.5 | 70.0 | 56.9 | 55.8 | 47.8 | 41.5 | 53.1 |
| LLaVA-Video-7B-Qwen2 | 29.9 | 1.5 | 53.0 | 19.3 | 39.1 | 33.8 | 38.8 | 30.8 |
| LLaVA-Video-72B-Qwen2 | 39.3 | 32.0 | 57.7 | 29.8 | 40.1 | 25.9 | 45.0 | 38.5 |

*Table 13.* Evaluation results on ReVSI under the 16-frame sampling setting. Note that the 16-frame setting doesn't have room size estimation and route planning tasks due to insufficient global scene context.

| Method | Numerical Question | | | Multiple-Choice Question | | Avg. |
|---|---|---|---|---|---|---|
| | Obj. Cnt. | Abs. Dist. | Obj. Size | Rel. Dist. | Rel. Dir. | |
| *Baseline* | | | | | | |
| Chance (Random) | - | - | - | 24.7 | 28.2 | - |
| Chance (Frequency) | 51.2 | 40.3 | 16.6 | 28.0 | 33.3 | 33.9 |
| *Open-Source Models* | | | | | | |
| Qwen3-VL-8B-Instruct | 33.5 | 38.8 | 65.7 | 44.2 | 36.5 | 43.7 |
| Qwen3-VL-32B-Instruct | 38.5 | 49.1 | 67.4 | 41.7 | 34.2 | 46.2 |
| InternVL3.5-8B | 43.5 | 40.3 | 63.5 | 42.8 | 35.4 | 45.1 |
| InternVL3.5-38B | 41.3 | 39.1 | 67.9 | 51.1 | 44.4 | 48.8 |
| LLaVA-Video-7B-Qwen2 | 31.1 | 1.7 | 51.8 | 37.2 | 33.5 | 31.1 |
| LLaVA-Video-72B-Qwen2 | 35.4 | 37.4 | 56.3 | 39.6 | 24.1 | 38.6 |

*Table 14.* Performance breakdown on ReVSI across task-specific metric splits, including single vs. multiple object counting and room size estimation, closest vs. farthest relative distance, and forward vs. backward relative direction.

| Method | Frames | Object Counting | | Room Size Est. | | Relative Distance | | Relative Direction | |
|---|---|---|---|---|---|---|---|---|---|
| | | Single | Multiple | Single | Multiple | Closest | Farthest | Forward | Backward |
| *Proprietary Models (API)* | | | | | | | | | |
| GPT-5.2 | 64 | 68.6 | 43.8 | 65.3 | 60.8 | 50.0 | 46.8 | 32.0 | 37.8 |
| Gemini 3 Flash | 1 FPS | 63.8 | 67.7 | 52.7 | 52.9 | 66.0 | 63.2 | 47.3 | 48.5 |
| Gemini 3 Pro | 1 FPS | 58.3 | 61.9 | 57.1 | 46.8 | 73.0 | 63.2 | 55.7 | 56.4 |
| *Open-Source Models* | | | | | | | | | |
| Qwen3-VL-8B-Instruct | 64 | 62.9 | 18.0 | 44.4 | 45.8 | 67.2 | 47.0 | 48.6 | 30.3 |
| Qwen3-VL-32B-Instruct | 64 | 62.4 | 31.4 | 44.9 | 66.6 | 68.2 | 39.3 | 50.0 | 17.9 |
| InternVL3.5-8B | 64 | 62.5 | 24.2 | 54.9 | 40.2 | 54.7 | 35.3 | 45.8 | 27.6 |
| InternVL3.5-38B | 64 | 67.6 | 19.9 | 59.1 | 57.8 | 66.6 | 48.2 | 66.2 | 25.6 |
| LLaVA-Video-7B-Qwen2 | 64 | 50.2 | 12.3 | 11.2 | 22.1 | 44.6 | 32.0 | 37.4 | 29.2 |
| LLaVA-Video-72B-Qwen2 | 64 | 52.1 | 28.2 | 31.6 | 24.2 | 42.4 | 36.8 | 24.3 | 25.3 |
| *Fine-Tuned Models* | | | | | | | | | |
| Cambrian-S-7B | 128 | 61.8 | 35.0 | 37.0 | 56.3 | 67.4 | 6.7 | 77.0 | 20.0 |
| VST-7B-SFT | 4 FPS | 60.2 | 10.5 | 40.4 | 54.0 | 61.5 | 36.9 | 51.2 | 22.5 |
| SpaceR-7B (SG-RLVR) | 32 | 45.3 | 16.1 | 18.4 | 18.7 | 22.8 | 22.9 | 44.2 | 24.8 |
| Spatial-MLLM-4B-135k | 16 | 49.4 | 32.1 | – | – | 45.8 | 18.7 | 55.0 | 19.9 |
| Spatial-MLLM-4B-820k | 16 | 52.5 | 30.4 | – | – | 44.5 | 17.0 | 51.5 | 26.9 |
| VLM3R-7B | 32 | 58.4 | 24.8 | 48.7 | 56.4 | 67.3 | 25.6 | 84.3 | 14.6 |

