# OpenReview forum: "ReVSI: Rebuilding Visual Spatial Intelligence Evaluation for Accurate Assessment of VLM 3D Reasoning"
_ICML.cc/2026/Conference — ICML 2026 regular_

### Official Review · Reviewer_U1Dn · 2026-03-03

**Soundness:** 4
**Presentation:** 3
**Significance:** 3
**Originality:** 3
**Overall Recommendation:** 4
**Confidence:** 5

**Summary:**

This paper presents ReVSI, a refined benchmark that corrects annotation errors in VSI-Bench and generates a more comprehensive set of questions with a more balanced answer distribution. To accommodate the varying input frame constraints of different models, this work introduces a frame-budget-adaptive benchmarking protocol that scales to different frame count limits while ensuring that all questions remain answerable. Additionally, this work designs a dummy-video setting to examine models' reliance on visual evidence, thereby assessing the degree of hallucination exhibited by the models.

**Compliance With Llm Reviewing Policy:**

Affirmed.

**Final Justification:**

Most of my concerns have been addressed; I maintain my recommendation.

**Key Questions For Authors:**

1. Why does Gemini 3 Flash perform much better than Gemini 3 Pro on the Abs. Dist. metric in Table 3? This seems counterintuitive. Could this indicate certain shortcomings in the MRA metric?
2. In Lines 382–383, the paper states that InternVL3.5 is easily misled by dummy videos that retain scene context but lack objects, and consistently predicts 2 on black videos. However, since InternVL 3.5 is evaluated in a zero-shot setting, why would it produce a constant prediction of 2? Such behavior seems more characteristic of overfitting in a fine-tuned model. Could this be a typo or a misstatement?

**Limitations:**

yes

**Strengths And Weaknesses:**

Strengths
1. The paper accurately identifies two fundamental flaws in VSI-Bench: annotation-to-video ground-truth drift and scene observability mismatch, with both arguments substantiated through quantitative evidence. For instance, Figure 2 and Table 7 systematically demonstrate a substantial degradation in question answerability and annotation accuracy in VSI-Bench, providing compelling empirical support for the authors' claims.
2. The re-annotation effort is substantial in scale, encompassing 5 scene datasets, 413 scenes, 5,436 object instances, and 466 open-vocabulary labels, along with the development of a complete web-based annotation toolchain.
3. The extensive experiments presented in this paper reveal the genuine spatial reasoning capabilities of various models. In particular, the dummy-video experiments uncover hallucination behavior in InternVL3.5, and demonstrate that fine-tuned models exhibit significant overfitting to VSI-Bench.

Weakness
1. Although Table 3 evaluates different models under their respective frame budgets, it lacks a controlled comparison in which all models are assessed under a unified frame count setting. For instance, the Gemini 3, originally evaluated under a 1 fps frame budget, could also be tested under the 32-frame setting to enable a more direct comparison with GPT-5.2.
2. The entries corresponding to Qwen2.5-7B-Instruct + SigLIP2 in Table 4 are entirely missing, rendering this row uninformative and of no referential value.

---

> ### Author Rebuttal · Authors · 2026-03-31
>
> We appreciate the constructive feedback and the recognition of our empirical evidence on annotation drift and observability mismatch. We also thank the reviewer for highlighting the value of our web-based annotation toolchain and the actionable insights from the dummy-video experiments. We address the questions below.
>
> &nbsp;
> ## Weakness 1: Evaluations under a unified frame count
> Our use of 1 FPS for Gemini 3 follows VSI-Bench, ensuring consistency with prior work. Moreover, the official Gemini API supports video input via frame rate (FPS) rather than a fixed number of frames, making 1 FPS the most natural configuration.
>
> To enable a controlled comparison, we additionally evaluate Gemini 3 Flash and Pro under a 32-frame setting by uniformly pre-sampling 32 frames per video and providing them as image inputs:
> | |Frames|Obj. Cnt.|Abs. Dist.|Obj. Size|Room Size|Rel. Dist.|Rel. Dir.|Route Plan|Avg.|
> |-|-|-|-|-|-|-|-|-|-|
> |Gemini 3 Flash|1 FPS|62.4|31.5|76.4|49.1|35.2|42.9|42.0|48.5|
> |Gemini 3 Flash|32|56.5|42.6|77.4|59.8|37.0|44.0|52.0|52.8|
> |Gemini 3 Pro|1 FPS|72.1|20.6|75.7|55.5|36.0|63.4|50.0|53.2|
> |Gemini 3 Pro|32|64.0|25.8|74.9|58.9|43.0|56.0|58.0|54.4|
>
> We observe that Gemini 3 achieves better performance under the 32-frame setting on some tasks. We hypothesize that increased visual tokens make reasoning more challenging. A similar trend is also reported in VSI-Bench’s paper (`Figure 11`), where some model performance degrades as frame count increases. This suggests an important direction for future work: enabling VLMs to focus on informative key frames in long videos, rather than uniformly attending to all frames.
>
> We also refer to `Tables 12 and 13` in the paper for results under 16- and 32-frame settings for other models.
>
> &nbsp;
> ## Weakness 2: Qwen2.5-7B-Instruct + SigLIP2 results in Table 4
> Qwen2.5-7B-Instruct + SigLIP2 is the base model used in Cambrian-S, where a Qwen language model is combined with a SigLIP2 visual encoder. Prior to fine-tuning, the embeddings from the two modules are not aligned, so the model is not directly usable for inference or to produce meaningful outputs. The original Cambrian-S paper also does not report results for this base model for the same reason. **Therefore, this entry is intentionally left without scores rather than missing.** We will clarify this in the final draft to avoid confusion.
>
> &nbsp;
> ## Question 1: Gemini 3 Flash vs. Gemini 3 Pro performance in Table 3
> We thank the reviewer for the insightful observation. We believe this behavior is plausible and does not indicate a shortcoming of the MRA metric.
>
> First, Gemini 3 Pro outperforms Flash on most metrics, suggesting that the overall trend remains consistent. Second, similar inversions have also been observed in official Gemini 3 reports, where Gemini 3 Flash outperforms Gemini 3 Pro on SWE-bench Verified, MMMU-Pro, and ARC-AGI-2, which are evaluated with different metrics. These results indicate that **performance does not strictly scale with model size**, and smaller or more efficient variants can outperform larger ones on specific tasks.
>
> Regarding the MRA metric, we follow the exact definition and parameter settings from prior work VSI-Bench, for fair comparison. As defined in `Equation 1`, **MRA does not lead to inconsistent model ranking—relative performance differences between models are stable and comparable.**
>
> &nbsp;
> ## Question 2: InternVL3.5 behavior
> This is not a typo or misstatement, we indeed observe this behavior consistently in the zero-shot evaluation of InternVL3.5. To substantiate this, we provide the detailed prediction distributions over 997 object counting questions under both *Dummy Video* and *Black Video* settings, numbers on the left denote predicted answers, and bar lengths indicate their proportions:
> ```text
> InternVL3.5-8B (Black Video)
>   0 | ████  17.65%
>   1 | █  1.71%
>   2 | ████████████████  80.64%
> ```
>
> ```text
> InternVL3.5-8B (Dummy Video)
>   0 | ███  14.74%
>   1 | ███  14.14%
>   2 | ██████████████  69.61%
>   3 | ▏1.20%
>   4 | ▏0.30%
> ```
>
> ```text
> InternVL3.5-38B (Black Video)
>   0 | ▏1.20%
>   1 | ████████████████  61.79%
>   2 | ███████████  36.71%
>   4 | ▏0.30%
> ```
>
> ```text
> InternVL3.5-38B (Dummy Video)
>   0 | ██  9.13%
>   1 | █████████  36.61%
>   2 | ████████████  52.66%
>   3 | ▏0.80%
>   4 | ▏0.60%
>   5 | ▏0.10%
>  10 | ▏0.10%
> ```
>
> These results show that the answer “2” dominates InternVL3.5’s predictions in most cases. This suggests that InternVL3.5 tends to rely on question priors rather than grounding its responses in actual visual input when performing object counting. This behavior could stem from earlier stages of the training pipeline, including data contamination in pretraining, imbalanced answer distributions, or pretraining strategies that lead the model to underutilize visual inputs. Thus, such behavior is not limited to fine-tuned models.

---

> > ### Author Rebuttal · Reviewer_U1Dn · 2026-04-01
> >
> > My concerns have been adequately addressed.

---

### Official Review · Reviewer_59Co · 2026-03-11

**Soundness:** 3
**Presentation:** 4
**Significance:** 3
**Originality:** 2
**Overall Recommendation:** 4
**Confidence:** 3

**Summary:**

This paper systematically investigates fundamental validity issues in existing visual spatial intelligence benchmark, VSI-Bench, and proposes ReVSI as an improved benchmark. The paper identifies two critical problems in VSI-Bench: (1) annotation-to-video ground-truth drift arising from using point-cloud-based 3D annotations for video-based evaluation, and (2) scene-observability mismatch where evaluations assume full-scene access while VLMs operate on sparsely sampled frames. To address these issues, ReVSI re-annotates 413 scenes across 5 datasets with professional 3D visualization tools, regenerates QA pairs with rigorous bias mitigation and human verification, and introduces frame-aware evaluation protocols across multiple frame budgets (16/32/64/all). The paper evaluates on different VLMs (proprietary and open-source) and systematically improves the accuracy and usability of the dataset.

**Compliance With Llm Reviewing Policy:**

Affirmed.

**Final Justification:**

The authors have resolved most of my concerns. I thus maintain my recommendation.

**Key Questions For Authors:**

Please refer to the weakness

**Limitations:**

yes

**Strengths And Weaknesses:**

### Strengths

1. Clear and intuitive issues identification. The paper addresses genuine validity issues in existing spatial intelligence benchmarks that have received insufficient attention. The systematic analysis of annotation quality (Figure 2) and frame-sampling effects (Figure 3) provides compelling empirical evidence that current evaluations are fundamentally flawed. This work could have significant impact on how VLM evaluations are conducted in the community.

2. Comprehensive benchmark quality improvements. The manual re-annotation effort is substantial and well-executed. Re-annotating 413 scenes with professional 3D tools, expanding object labels from 65 to 466 categories with open-vocabulary support, and increasing object instances from 3,185 to 5,436 represents significant effort. The comparison in Table 1 and detailed error analysis in Figure 2 demonstrate meaningful improvements in annotation accuracy.

3. Surprising and actionable insights from the evaluation. The paper reveals important insights: proprietary models are under-assessed by VSI-Bench on counting tasks (Table 3), fine-tuned models show minimal gains over baselines on ReVSI despite large improvements on VSI-Bench (Table 4), and specialized models exhibit catastrophic hallucination on dummy videos (Table 5). These findings challenge prior conclusions and provide concrete evidence of evaluation validity issues.

### Weaknesses

**Major**

1. The MRA metric (Eq. 1) uses relative error, which may systematically favor longer-range distances by being more tolerant at larger ground-truth values. Figure 7 illustrates this, but the paper frames this as evidence that ReVSI "better reflects true spatial reasoning" rather than acknowledging that the metric change itself contributes to score differences. A more neutral analysis would compare results using identical metrics on both benchmarks.

2. The paper modifies question templates from VSI-Bench (e.g., "How many objects are in this room?" to "How many objects are in the scene?") and redesigns answer distributions (Figure 5) substantially. However, the modification makes it difficult to isolate whether performance differences stem from annotation quality improvements or question design changes. A side-by-side comparison using original questions on re-annotated data would strengthen claims.

3. The paper mentions "all annotation tasks were performed directly by authors" (Section 4.1), which raises concerns about potential systematic bias. Additionally, the use of GPT-5.2 for "verification" of object labels is not clearly explained—how does this verification work, and could it introduce bias?

**Minor**

1. Missing analysis of task-specific failure modes. While the paper identifies hallucination as a common failure mode, it lacks detailed analysis of which specific task categories (object counting, size estimation, relative distance, etc.) are most affected by annotation quality improvements versus frame sampling effects. Task-specific breakdowns would provide more actionable insights for model developers.

2. Dummy video construction details underspecified. The paper removes frames "containing any object referenced by the associated question" (Section 5.2), but how are partial or ambiguous cases handled? If an object appears only partially in a frame, is that frame removed? This affects the validity of the dummy-video diagnostics.

---

> ### Author Rebuttal · Authors · 2026-03-31
>
> We thank the reviewer for recognizing our identification of validity issues and benchmark improvements. We address the insightful questions below.
>
> &nbsp;
> ## Major W1
> As stated in `Sec. 6.1`, **we use the same MRA metric and parameters for all exps as VSI-Bench with no modifications.** All comparisons follow identical protocols. `Fig. 7` is intended to analyze the properties of the metric, rather than to suggest any difference in metric setup.
>
> Our claim that ReVSI better reflects spatial reasoning stems from data distribution. As shown in `Fig. 5`, VSI-Bench has a large portion of short-range (0–2 m) distance queries solvable from single frames, while ReVSI emphasizes long-range queries, encouraging spatial reasoning beyond single-frame perception.
>
> &nbsp;
> ## Major W2
> We categorize our template changes into (a) new template variants and (b) modifications to existing VSI-Bench templates.
>
> For (a), `Tab. 14` already reports results using unchanged VSI-Bench templates (e.g., room size (single), rel. dist. (closest), rel. dir. (forward)).
>
> For (b), minor modifications are made for:
> - **Obj. counting:** VSI-Bench uses “in this room,” but videos often span multiple rooms and their GT counts the whole scene, leading to misalignment (`Sec. 4.2`).
> - **Obj. size est.:** we prepend “Based on visual evidence from the video” to encourage grounding in the input and reduce reliance on priors.
>
> We further compare both templates on the two tasks using ReVSI annotations:
> | |Obj. Cnt. (VSI-Bench Template)|Obj. Cnt. (ReVSI Template)|Obj. Size (VSI-Bench Template)|Obj. Size (ReVSI Template)|
> |-|-|-|-|-|
> |Gemini 3 Pro|57.2|62.2|81.4|83.9|
> |Qwen3-VL-32B|59.9|60.0|72.4|72.7|
> |InternVL3.5-38B|65.7|65.8|71.5|71.5|
>
> We observe template modifications mainly affect the stronger proprietary model (Gemini 3 Pro), with negligible impact on others. We attribute this to the fact that stronger models are better able to follow rigorous instructions. This further indicates our revised templates improve the evaluation upper bound and provide a more rigorous protocol for future models.
>
> &nbsp;
> ## Major W3
> Compared to prior work, we reduce rather than introduce bias. We categorize our annotations into three components: (1) 3D object boxes and object names, (2) object visibility in video, and (3) room area.
>
> (2) and (3) have minimal systematic bias since object visibility is a binary decision (visible or not) based on direct visual evidence, and room area follows widely accepted architectural conventions aligned with common sense. Thus, the primary source of potential bias is object naming. VSI-Bench directly inherits labels from existing datasets (e.g., ScanNet, ScanNet++, ARKitScenes). Notably, ScanNet’s final annotations were corrected by a single author, while ScanNet++ annotations were produced by only 2–5 students.
>
> In contrast, we reannotated object labels using human–LLM cross-verification strategy to improve label accuracy and diversity. Annotators first assign open-vocabulary labels based on visual evidence. GPT-5.2 is then used solely for verification (e.g., correcting errors or suggesting refinements), with humans making final decisions. Specifically, annotators manually select 1–3 views with clear object visibility, manually crop tight object-centric regions, and query GPT-5.2 with prompt *These are images of an object, what is the name of the object?*. Outputs are compared with annotators' label. If no clear agreement can be reached, the object is discarded.
>
> &nbsp;
> ## Minor W1
> We have isolated the frame sampling issue from both data (`Fig. 3`, `Tab. 7`) and model performance perspectives (`Tab. 3, 12 and 13`). `Fig. 3` studies answerability and correctness across frame budgets independent of annotation quality, while `Tab. 3, 12, 13` compare identical models under 16/32/64-frames settings to assess performance sensitivity. Notably, comparisons are conducted after removing unanswerable questions under each setting, since sparse sampling eliminates necessary visual evidence (`Fig. 3`, `Tab. 7`), a strictly identical question set across different frame budgets is not meaningful.
>
> To assess annotation quality, we decouple its effect from template changes (see response to *Major Weakness 2*). We then compare `Tab. 14` (room size (single), rel. dist. (closest), rel. dir. (forward), with VSI-Bench templates) with the corresponding VSI-Bench results in `Tab. 3`. This isolates the impact of re-annotation on model performance.
>
> &nbsp;
> ## Minor W2
> For each question, we remove all frames where the queried object has ≥1 visible pixel (including partial visibility) to ensure valid dummy-video diagnostics. We obtain per-frame 2D object masks by rasterizing scene meshes from GT camera poses, and discard any frame containing the object pixels. We then uniformly sample 16 frames (repeating if needed), followed by an additional round of human verification to confirm complete object invisibility. These details will be clarified in the final draft.

---

> > ### Author Rebuttal · Reviewer_59Co · 2026-04-02
> >
> > The authors have resolved most of my concerns. I thus maintain my recommendation.

---

### Official Review · Reviewer_cfrf · 2026-03-12

**Soundness:** 2
**Presentation:** 3
**Significance:** 3
**Originality:** 2
**Overall Recommendation:** 4
**Confidence:** 4

**Summary:**

This paper argues that the widely used spatial intelligence benchmark VSI-Bench can be systematically invalid due to annotation errors stemming from point-cloud–derived labels and unrealistic assumptions about full-scene access. To address this issue, the authors introduce ReVSI, a benchmark that re-annotates 413 scenes from five datasets and regenerates QA pairs to ensure correctness and answerability under the actual inputs available to VLMs. The benchmark also provides multiple frame-budget settings and object visibility metadata to enable controlled evaluation. Experiments show that ReVSI reveals systematic failure modes in VLM spatial reasoning that are often obscured by prior benchmarks.

**Compliance With Llm Reviewing Policy:**

Affirmed.

**Final Justification:**

I appreciate the authors' detailed responses, which have resolved the majority of my concerns. I am pleased to maintain my positive recommendation.

**Key Questions For Authors:**

1. You mention that Object Appearance Order task is excluded because it primarily assesses temporal reasoning and involves ambiguous "appearance" definitions for boundary objects. Could you provide specific visual examples or edge cases where "appearance" becomes ill-defined? How would you distinguish a task that is "purely" 3D spatial from one that is "purely" temporal in a video context?
2. Could you elaborate on why object counting is classified as a 3D spatial reasoning task? While you mention that distinguishing repeated observations of the same object in long videos requires 3D reasoning, how does the ReVSI dataset specifically ensure that a model must use 3D cues (like spatial localization) rather than just simple 2D temporal tracking to arrive at the correct count?
3. Given the "costly expert-level human annotation" mentioned in your limitations, have you considered or tested any semi-automated pipelines to assist in the re-annotation process?

**Limitations:**

yes

**Strengths And Weaknesses:**

**Strengths:**
1. This paper does a comprehensive analysis of validity pitfalls in current VSI evaluation. Specifically, it identifies two critical pitfalls: Annotation-to-video ground-truth drift (errors in repurposed point-cloud data) and Scene-observability mismatch (questions that are unanswerable given sparse frame sampling).
2. Unlike prior benchmarks that use a static ground truth, ReVSI introduces a dynamic, frame-budgeted protocol (16/32/64/all frames). This ensures that the ground truth is strictly consistent with the visual evidence actually available to the model.


**Weaknesses:**
1. While ensuring high data quality, the reliance on expert-level manual 3D annotation creates a significant hurdle for scaling the benchmark.
2. While the paper argues for the necessity of 3D reasoning, the distinction between 2D perception and 3D spatial reasoning in certain tasks (like object counting) remains thin. Furthermore, the justification for excluding the Object Appearance Order task could be more robustly supported with quantitative examples of "appearance" ambiguity.

---

> ### Author Rebuttal · Authors · 2026-03-31
>
> We thank the reviewer for the positive evaluation of our work. We appreciate the recognition of our comprehensive analysis of validity pitfalls and the acknowledgment of our dynamic frame-budgeted protocol as an important step toward ensuring visual evidence consistency. We address the insightful questions below:
>
> &nbsp;
> ## Weakness 1: Scalability.
> We agree expert annotation limits scalability, and we’ve discussed it in the paper as future work. However, **this limitation mainly affects training data generation, rather than ReVSI's role as an evaluation dataset.** While large-scale data is critical for training, evaluation requires a balance between scale and annotation fidelity to ensure reliable assessment. Moreover, evaluation cost (in terms of tokens) is also an important consideration.
>
> ReVSI already exceeds the prior work VSI-Bench in scale (`Tab. 1`), and we believe it is sufficient for evaluation purposes.
>
> &nbsp;
> ## Weakness 2 & Question 1 / 2: What constitutes a 3D spatial task?
> A task requires 3D spatial reasoning if it necessitates maintaining object identity across **non-adjacent** views, which involves reasoning about object locations and camera motion over time. In contrast, tasks solvable via frame-wise or short-range temporal cues do not require 3D reasoning.
>
> Under this definition, **appearance order is not a 3D task**. Prior work (e.g., VLM-3R [1], see their GitHub issue `#25`) also excludes this task, indicating this is a known limitation. To further clarify, we provide representative edge cases where “appearance” becomes ill-defined. In [example 1](https://github.com/anonymous-2026-03/anonymous/blob/main/appearance_order_example_1.pdf), a toilet paper roll is partially visible as early as `Frame 15` but is extremely small and occluded. It only becomes clearly identifiable in `Frame 1018`. And [example 2](https://github.com/anonymous-2026-03/anonymous/blob/main/appearance_order_example_2.pdf) shows a similar case.
>
> In contrast, **object counting is inherently a 3D task**. Prior work Cambrian-S [2] also emphasizes counting as a key component of 3D spatial intelligence, supporting our categorization. Counting requires reasoning about global 3D layout and camera trajectory to determine whether an observed object is new or previously seen (e.g., when the camera revisits the same location, which is common in scene datasets). For example, in a classroom with many similar chairs in rows, each frame shows only a local view, and different rows appear indistinguishable. A model relying on 2D tracking cannot tell whether a newly observed row is new or previously seen (as camera revisits are common). Accurate counting therefore requires reasoning about global 3D layout and camera trajectory to avoid double-counting.
>
> &nbsp;
> ## Question 3: Potentials of semi-automated data generation pipelines.
> We thank the reviewer for this important question. We explored potential semi-automated pipelines using state-of-the-art models, including 3D detection (VDETR+DEST [3]), 3D segmentation (Sonata [4]), 2D detection (Grounding DINO [5]), and rasterization-based object visibility estimation from 3D annotations and camera poses. However, these methods degrade annotation quality for three key reasons:
> 1. **Mismatch between 3D geometry and video:**
> Existing datasets have noisy 3D geometry and annotations, especially those captured with commodity sensors, leading to inaccurate geometry and camera poses. As a result, both ground-truth annotations and model predictions are often misaligned with raw video frames (e.g., `Teaser (a)` and `Fig. 10`). Even SoTA 3D detection models perform poorly (e.g., VDETR+DEST achieves only 67.9 AP50 on ScanNet v2). This undermines the reliability of video-based QA construction, a core issue in VSI Bench, and necessitates expert reannotation.
> 2. **Limitations of 2D per-frame methods:**
> Both VSI-Bench and ReVSI require multi-frame 3D annotation, which cannot be reliably captured by 2D detection or independent frame-wise predictions due to lack of long-range cross-view consistency.
> 3. **Failure of rasterization-based visibility estimation:**
> We attempted to compute per-frame object visibility by rasterizing 3D objects using GT camera trajectories. However, due to geometry holes and pose errors, rendering often produces incorrect projections, leading to unreliable object visibility.
>
> Based on these limitations, we adopt a fully manual annotation pipeline to ensure high fidelity.
>
> ---
>
> References:
>
> [1] Fan et al., VLM-3R: Vision-language models augmented with instruction-aligned 3D reconstruction., CVPR 2026
>
> [2] Yang et al., Towards Spatial Supersensing in Video., ICLR 2026.
>
> [3] Wang et al., State Space Model Meets Transformer: A New Paradigm for 3D Object Detection., ICLR 2025
>
> [4] Wu et al., Sonata: Self-supervised learning of reliable point representations., CVPR 2025
>
> [5] Liu et al., Grounding DINO: Marrying DINO with grounded pre-training for open-set object detection., ECCV 2024

---

> > ### Author Rebuttal · Reviewer_cfrf · 2026-04-04
> >
> > Really appreciate the response! I would like to maintain the current positive score.

---

### Official Review · Reviewer_pU38 · 2026-03-17

**Soundness:** 3
**Presentation:** 2
**Significance:** 3
**Originality:** 3
**Overall Recommendation:** 4
**Confidence:** 5

**Summary:**

This paper re-investigates the VSI-benchmark, which then finds there are several misalignment issues in annotation correctness and bias, such as missing geometry in sampled video frames, wrong object labels, and ambiguous space. To address such issues, this paper then re-annotates object labels and scene geometry using expert annotation tools, re-generates QA pairs with more fine-grained captions, and re-defines frame-budgeted evaluation.  They curate a rigorous benchmark that enforces frame-budget–aware evaluation, visibility-consistent question generation, and systematic human verification across spatial reasoning tasks. Moreover, this paper designs diagnostic tools to provide a reliable way for analyzing, comparing, and developing video-language models for 3D spatial intelligence.

**Compliance With Llm Reviewing Policy:**

Affirmed.

**Final Justification:**

I appreciate the response from the authors and most of my concerns have been addressed. I would like to maintain my initial score.

**Key Questions For Authors:**

This paper should clarify and explain in more detail tables 3 and 4.

And the video frames result in vulnerable 3D spatial intelligence performances.

The revisited benchmark sounds not that enough scales to cover diverse scenes to better demonstrate the models' generalization.

**Limitations:**

Please see above.

**Strengths And Weaknesses:**

Strengths:
1. This paper systematically revisits the evaluation issues on the VSI-benchmark for 3D spatial intelligence, which takes great effort to investigate and analyze.

2. The proposed claims that raise the issues on the VSI-benchmark sound reasonable.

3. The important problem of the video frames sampling presents a significant challenge for video-based 3D spatial intelligence, while humans can continually perceive the video-stream, but models cannot due to the high computational cost for numerous video frames.



Weaknesses:
1. Regarding Table 3, I am confused that the revisited benchmark ReVSI leads to inferior results compared with the noisy VSI-benchmark.

2. Regarding Table 4, the input frames for some SoTAs, like Spatial-MLLM-4B-135k, VLM3R-7B, are missing; the authors should clarify this.

3. This paper should also do evaluations across various video frames on the same model, to know Frame-budgeted and Visibility-guided controlled evaluation fairness.

---

> ### Author Rebuttal · Authors · 2026-03-31
>
> We thank the reviewer for the positive assessment, particularly for recognizing the systematic effort invested in analyzing this benchmark. We are encouraged that our claims are viewed as reasonable and that the reviewer agrees on the critical challenge of video frame sampling. We address the questions and concerns as follows:
>
> &nbsp;
> ## Weakness 1 & Question 1: Clarification of lower model scores on ReVSI (Table 3)
> We would like to clarify that **higher-quality benchmarks do not imply higher model scores**. ReVSI is designed to more faithfully reflect models’ true 3D spatial reasoning ability by removing annotation noise, enforcing visibility consistency, and aligning evaluation with the actual input frames. The observed performance drop indicates that the prior benchmark VSI-Bench may overestimate model capability due to annotation artifacts and frame misalignment. In contrast, ReVSI exposes these shortcomings.
>
> Moreover, the lower performance on ReVSI is not uniform across all models. Some models, particularly proprietary models, achieve higher performance on certain tasks (e.g., numerical questions) under ReVSI. Our ablation studies in `Table 5` further indicate that one contributing factor is their lower hallucination rate.
>
> Overall, these findings suggest that **ReVSI provides a more discriminative and reliable evaluation, rather than systematically increasing or decreasing benchmark scores**.
>
> &nbsp;
> ## Weakness 2 & Question 1: Clarification of frame numbers (Table 4)
> The input video frame settings for models such as Spatial-MLLM and VLM3R are inherited from their corresponding base models, where the frame numbers are explicitly listed. We **apply identical frame settings to each base model and its fine-tuned models, and omit repeated entries in the table for brevity**. We thank the reviewer for pointing out the confusion, we will explicitly include the input frame numbers in the final draft for every row to improve clarity.
>
> &nbsp;
> ## Weakness 3 & Question 1: Evaluation across different frame budgets
> We agree that analyzing performance under varying frame budgets is important. We have already included such evaluations in:
> - `Table 1`: 64-frames setting (main paper)
> - `Tables 12 & 13`: 32-frames and 16-frames settings (appendix)
>
> Together, these results compare model robustness under different frame budgets and validate the fairness of our frame-budgeted evaluation protocol. We will highlight these results more clearly in the final draft.
>
> &nbsp;
> ## Question 2: Dataset scale and diversity
> ReVSI expands the dataset scale along multiple dimensions compared to VSI-Bench:
> - **Scene data sources:** increased from 3 to 5 datasets (ScanNet v2, ScanNet++, ARKitScenes, 3RScan, MultiScan), including diverse real-world scans from North America, China, and Europe.
> - **Scale:** increased number of scenes, objects, open-vocabulary labels, and QA pairs (see `Table 1` and `Figure 4`), which have also been recognized by reviewers `U1Dn` and `59Co` as substantial in annotation effort and scale.
>
> We agree that scale is important. However, **the primary goal of ReVSI is to improve evaluation fidelity and reliability, rather than to maximize dataset size**. Given the evaluation cost of video-language models and the scale of existing benchmarks, we believe ReVSI achieves a strong balance between quality and scale, and provides a more trustworthy testbed for future research.

---

> > ### Author Rebuttal · Reviewer_pU38 · 2026-04-06
> >
> > I appreciate the response from the authors and most of my concerns have been addressed. I would like to maintain my initial score.

---

### Decision · Program_Chairs · 2026-04-30

**Decision:**

Accept (regular)

**Comment:**

This paper was evaluated by four expert reviewers, all of whom recommended Weak Accept. Taking into account the overall reviewer feedback and the author's response, the recommendation is to accept the paper. The reviews nevertheless raised several constructive concerns that merit attention in the camera-ready version, in particular the need for stronger experimental comparisons and ablation studies, as highlighted by Reviewers 59Co and U1Dn, as well as clearer presentation and framing of the paper, as noted by Reviewers pU38 and cfrf. The authors are encouraged to address these points to the extent possible in the final version.